# Real-Aware Residual Model Merging for Deepfake Detection

## Abstract

Deepfake generators evolve quickly, making exhaustive data collection and repeated retraining impractical. We argue that model merging is a natural fit for deepfake detection: unlike generic multi-task settings with disjoint labels, deepfake specialists share the same binary decision and differ in generator-specific artifacts. Empirically, we show that simple weight averaging preserves Real representations while attenuating Fake-specific cues. Building upon these findings, we propose Real-aware Residual Model Merging ($R^2M$), a training-free parameter-space merging framework. $R^2M$ estimates a shared Real component via a low-rank factorization of task vectors, decomposes each specialist into a Real-aligned part and a Fake residual, denoises residuals with layerwise rank truncation, and aggregates them with per-task norm matching to prevent any single generator from dominating. A concise rationale explains why a simple head suffices: the Real component induces a common separation direction in feature space, while truncated residuals contribute only minor off-axis variations. Across in-distribution, cross-dataset, and unseen-dataset, $R^2M$ outperforms joint training and other merging baselines. Importantly, $R^2M$ is also composable: when a new forgery family appears, we fine-tune one specialist and re-merge, eliminating the need for retraining.

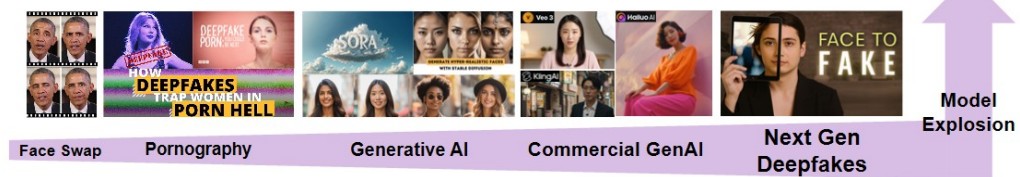

Figure 1: **Conceptual timeline of deepfakes**: all thumbnails are synthetic (fake), and the number of generative models grows explosively from early face swaps to commercial/unknown GenAI

## 1 Introduction

As sketched in Fig.1, deepfakes have progressed from simple face splicing to photorealistic synthesis powered by modern generative models (Li et al., 2019; Chen et al., 2020; Tolosana et al., 2020; Rombach et al., 2022a). Recent systems preserve identity while controlling lip movements and expressions, and high-quality content can now be produced by anyone through simple prompts on widely available generative services and APIs (Prajwal et al., 2020; Park & Owens, 2025). This accelerates the spread of both legitimate media and potentially harmful content, including financial fraud, copyright violations, and political disinformation, necessitating more reliable detection.

Because of rapid diversification, exhaustively collecting per-algorithm data and retraining is infeasible. Even if sufficient data were available, joint training on heterogeneous forgeries suffers from interference (Yu et al., 2020; Standley et al., 2020), while maintaining one specialist per generator is operationally costly (Shiohara & Yamasaki, 2022; Yan et al., 2024b). We therefore adopt a *model merging* approach: specialists are fine-tuned on their own data and then combined in parameter space to form a single detector, enabling rapid adaptation without retraining (Izmailov et al., 2018; Wortsman et al., 2022; Ilharco et al., 2022). To the best of our knowledge, model merging has not been systematically explored for deepfake detection. In addition to this structural motivation,

we provide empirical evidence that averaging specialist parameters consistently retains real features while attenuating generator-specific fake cues.

To ground this choice, we analyze how a weight-averaged (WA) model (Izmailov et al., 2018) compares to its specialists. We take per-forgery-family specialists finetuned from the same pretrained backbone and construct a WA model by averaging their parameters without retraining. For clarity, we denote the three forgery families in DF40 (Yan et al., 2024b) as FS (FaceSwap), FR (FaceReenactment), and EFS (Entire Face Synthesis). We measure the similarity between each specialist and the WA model on three data types: Real images, Own-Fake (fake images produced by that specialist's forgery family), and Other-Fake (fake images from the remaining families), using cosine similarity of pre-logit features.

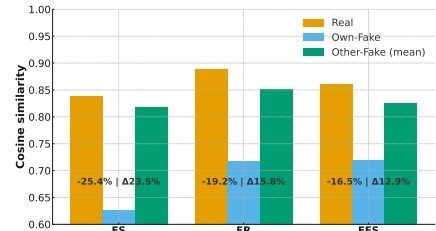

Figure 2: **Similarity between each specialist and the weight-averaged model (WA)** on Real, Own-Fake, and Other-Fake. FS, FR, and EFS denote *Face Swap*, *Face Reenactment*, and *Entire Face Synthesis*. Specialists are highly aligned with WA on Real, while Own-Fake shows a clear drop relative to both Real and Other-Fake, consistent with WA preserving shared Real structure and canceling generator-specific residuals.

The pattern is consistent across forgery families. On Real images, all specialists remain highly similar to the WA model, indicating that WA preserves the shared Real structure. On Own-Fake, similarity drops relative to both Real and Other-Fake, since strong generator-specific fake cues emerge and reduce alignment with WA. In contrast, Other-Fake carries weaker specialist-specific signals, so similarity to WA remains higher. Fig.2 visualizes this trend for FS, FR, and EFS, supporting the view that WA preserves shared Real structure while suppressing generator-specific Fake residuals. These observations motivate a domain-tailored merge strategy for deepfakes: retaining the Real component while recombining Fake-specific residuals.

Building on these observations, we tailor merging to the structure of deepfake detection. Unlike typical multi-task settings with disjoint label spaces, our specialists share the same binary label space (Real or Fake). In practice, cues for Real are stable across datasets, whereas cues for Fake are generator-specific and volatile. A suitable merging rule should therefore preserve the common structure among specialists while aggregating their complementary, generator-specific knowledge.

**Method overview.** We propose *Real-aware Residual Model Merging ($R^2M$)* (Fig. 3). We estimate a shared Real component through low-rank factorization (*e.g.*, SVD) of specialists' task vectors, treating the dominant directions as a core Real subspace. Each task vector is then decomposed into a Real-aligned part and a residual that captures generator-specific Fake evidence. We keep one shared Real component, denoise residuals with low-rank truncation, and merge the residuals with per-task norm matching to prevent

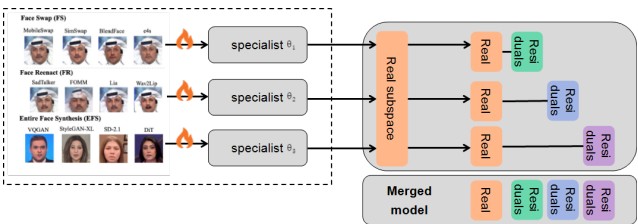

Figure 3: **Overview of Real-aware Residual Model Merging ($R^2M$).** Specialist detectors are fine-tuned independently. We then factorize their task vectors into a shared Real subspace and generator-specific residuals, keep a single Real core, and recombine denoised, norm-matched residuals into a merged detector.

any single generator from dominating the decision. The procedure is composable: integrating a new generator only requires finetuning its specialist and adding its residual to the merge. We detail the method in §3.

**Mechanistic intuition.** Uniform weight averaging tends to suppress generator-specific residuals and emphasize a shared Real component, consistent with our similarity probes. $R^2M$ makes this behavior explicit: it preserves the Real core and retains generator-specific Fake residuals, combining them at matched scales. From a local linear perspective around the pretrained model, the factorization recovers Real-aligned directions, while informative residual energy concentrates in a low-alignment subspace that carries generator-dependent artifacts. We present this rationale in §3.2 and demonstrate improvements in both seen-task retention and generalization to unseen forgeries in §4.

**Contributions.**

• We introduce **R²M**, a training-free recipe that preserves a shared Real component while composing denoised, norm-matched Fake residuals.

• We provide a concise rationale linking parameter-space factorization to feature-space geometry, explaining why weight averaging specializes to Real and why R²M decouples Real and Fake effects.

• We show consistent, reliable improvements over strong baselines. Furthermore, the merge is composable, enabling rapid incorporation of new forgery families.

## 2 RELATED WORKS

**Deepfake detection.** Deepfake technology has advanced rapidly over time. Early systems focused on simple local manipulations within the face region (fac, 2019; Li et al., 2020a; Liu et al., 2023b). The advent of GAN-based generators markedly improved synthesis quality (Choi et al., 2018; Thies et al., 2019; Richardson et al., 2021), followed by diffusion-based models that improved fidelity and enabled finer-grained control over generation (Rombach et al., 2022a). Beyond basic face replacement, talking-head and reenactment methods now preserve identity while naturally controlling facial expressions, emotions, speech content, lip synchronization, and head motion (Nirkin et al., 2019; Li et al., 2024; Mukhopadhyay et al., 2024; Guo et al., 2024). Commercial generative platforms such as Veo, Kling, and Wan [1] further lowered the barrier to producing high-quality synthetic videos. While the same techniques power legitimate creative media, they also facilitate realistic manipulations with significant privacy and safety risks.

On the detection side, prior work has evolved along several complementary approaches. Data augmentation via blending (Li et al., 2020b; Shiohara & Yamasaki, 2022; Lin et al., 2024) and frequency-domain analysis (Jeong et al., 2022; Tan et al., 2024; Zhou et al., 2024) are representative strategies. To cope with the continual emergence of new generators, recent studies emphasize generalization to *unseen* forgeries (Yan et al., 2023; Choi et al., 2024b; Cui et al., 2025). For example, Shiohara & Yamasaki (2022) generated pseudo-fakes through self-blending for training, and Yan et al. (2023) extracted common forgery features to improve transfer. Tan et al. (2024) leveraged high-frequency cues, while Lin et al. (2024) proposed temporally aware self-blending with curriculum learning. In parallel, Cui et al. (2025) improved efficiency with an adaptor-based architecture, and Sun et al. (2025) incorporated vision–language signals for detection. Yan et al. (2024a) further study generalizable AI-generated image detection by using SVD to construct orthogonal semantic and forgery subspaces in a vision foundation model, freezing the principal components and learning forgeries in the orthogonal residual subspace.

Despite these advances, robust generalization remains challenging under rapid diversification of generation methods and the proliferation of commercial tools. We therefore advocate a complementary perspective for deployment: *model merging* as a practical mechanism for deepfake detection. Rather than retraining an all-in-one detector whenever new forgeries appear, separately finetuned specialists can be combined into a single model in parameter space, enabling swift incorporation of new forgery families without full retraining.

**Model merging.** Model merging combines task-specific experts into a single model by operating directly in parameter space, typically without additional training. A basic approach is *weight averaging* (WA) (Izmailov et al., 2018), which averages parameters across experts. In the context of fine-tuning large pretrained models, WA underlies "model soups" and often improves accuracy and robustness without inference overhead (Wortsman et al., 2022). Beyond uniform averaging, *task arithmetic* represented each task by a *task vector* (the difference between fine-tuned and pretrained weights) and edits model behavior by adding or negating such vectors; combining multiple task vectors can yield multi-task capability (Ilharco et al., 2022). *TIES-Merging* addressed interference when merging by trimming small updates, resolving sign conflicts, and merging only sign-aligned parameters, achieving stronger multitask performance across modalities (Yadav et al., 2023). Recent work further revisits WA through the task-vector lens, showing that centering task vectors around the weight average and applying *low-rank* approximations to those vectors (CART) can substantially

---

[1]Veo : `https://deepmind.google/models/veo` Kling AI: `https://klingai.com`; Wan: `https://wan.video/`

improve merged performance by reducing cross-task interference (Choi et al., 2024a). Orthogonal directions include *Fisher-weighted averaging*, which weights parameters by local curvature when merging (Matena & Raffel, 2022). Recent studies also explore low-rank and interference-aware merging for parameter-efficient fine-tuning and cross-modal settings (Lee et al., 2025).

**Positioning of Our Work.** To the best of knowledge, model merging has *not* been systematically tailored to deepfake detection. Prior methods are largely task-agnostic, treating all task updates symmetrically. In contrast, our setting exhibits a structural asymmetry (shared Real vs. generator-specific Fake). We leverage this structure by explicitly preserving a shared Real component and recombining denoised, norm-matched Fake residuals, yielding a single detector that remains composable as new forgery families emerge. Our approach is complementary to training-time generalization methods : they aim to train a single detector with stronger cross-generator generalization, whereas we address how to *merge* multiple already-trained specialists in parameter space. In principle, specialists trained with such objectives could themselves be merged via $R^2M$.

## 3 METHOD

Prior model merging methods are commonly designed for multi-task settings with disjoint label spaces and focus on mitigating cross-task interference (Wortsman et al., 2022; Ilharco et al., 2022; Ainsworth et al., 2022; Yadav et al., 2023). Deepfake detection differs: specialists share the same Real vs. Fake label space; variation arises from generator-specific artifacts. We therefore separate what specialists have in common (a shared *Real* component) from what they learn differently (generator-specific *Fake residuals*) and recompose them into a single detector. Before aggregation, we perform per-task norm matching on residuals to equalize their energy, so that no single specialist dominates the merge. Our approach is a training-free, parameter-space merging method that preserves the strengths of specialists while improving generalization to unseen generators.

We denote the network parameters by $\theta \in \mathbb{R}^D$ and write $\theta_0$ for the *pretrained* weights. For each task $i \in [N] = \{1, \ldots, N\}$, *targeting detection of forgeries produced by a particular manipulation method*, let $\mathcal{D}_i \subset \mathcal{X} \times \{0, 1\}$ denote the dataset containing both real samples and the corresponding forgeries (labels $y \in \{0 \text{ (Real)}, 1 \text{ (Fake)}\}$). The specialist finetuned on task $i$ is denoted by $\theta_i$, and its *task vector* (Ilharco et al., 2022) is defined as $\tau_i := \theta_i - \theta_0$. We construct a single merged model $\theta_\star = \mathrm{Merge}(\theta_0, \{\theta_i\}_{i=1}^N)$ using the closed-form rule described in §3; with *no further training or gradient updates* performed during merging.

### 3.1 SVD-BASED DISENTANGLEMENT AND RANK-TRUNCATED DENOISING

**Task matrix, centering, and Real core.** The task vectors $\{\tau_i\}_{i=1}^N$ of specialists are stacked row-wise:

$$M = \begin{bmatrix} \tau_1^\top \\ \vdots \\ \tau_N^\top \end{bmatrix} \in \mathbb{R}^{N \times D}, \qquad \bar{\tau} = \frac{1}{N} \sum_{i=1}^N \tau_i. \tag{1}$$

Then, centering across parameters is performed using the all-ones vector $\mathbf{1} \in \mathbb{R}^N$:

$$M_c = M - \mathbf{1}\,\bar{\tau}^\top, \qquad M_c = U_c \Sigma_c V_c^\top. \tag{2}$$

Let $V_{c,k}$ denote the top-$k$ right singular vectors. We can define the Real projector and core:

$$\Pi_{\text{real}} := V_{c,k} V_{c,k}^\top, \qquad \tau_{\text{core}} := \Pi_{\text{real}} \bar{\tau} = \frac{1}{N} \sum_{i=1}^N \Pi_{\text{real}} \tau_i. \tag{3}$$

This extracts a low-rank component shared across specialists and aligned with Real.

**Residuals and layerwise rank-$r$ truncation.** We form mean-centered residuals as $\delta_i := \tau_i - \bar{\tau}$. For each attention and MLP block, we apply *layerwise SVD* to corresponding matrix slice of $\delta_i$ and keep only the top-$r$ singular components; yielding the truncated residual $\tilde{\delta}_i$. (Complete notation for truncated SVDs is provided in §A.1.)

**Across-task merge and final parameters (training-free).** Each truncated residual is normalized and rescaled to the mean residual norm to avoid dominance:

$$m_{\text{mean}} \;=\; \frac{1}{N}\sum_{i=1}^{N}\big\|\tilde{\delta}_i\big\|_2, \qquad \hat{\delta}_i \;=\; m_{\text{mean}}\,\frac{\tilde{\delta}_i}{\big\|\tilde{\delta}_i\big\|_2 + \varepsilon}. \tag{4}$$

Using uniform weights, we simply average the normalized residuals:

$$\tau_{\text{merge}}^{\text{res}} \;=\; \frac{1}{N}\sum_{i=1}^{N}\hat{\delta}_i. \tag{5}$$

The global residual scale is set relative to the Real core, after which we define the merged parameters:

$$\eta \;=\; \alpha\,\big\|\tau_{\text{core}}\big\|_2, \qquad \theta_{\text{R}^2\text{M}}(\alpha, r) \;=\; \theta_0 \;+\; \tau_{\text{core}} \;+\; \eta\,\tau_{\text{merge}}^{\text{res}}. \tag{6}$$

All operations are closed-form, with no additional training or gradient updates. The only tunable hyperparameters are $\alpha$ (global residual scale) and $r$ (per-layer SVD-$r$ truncation; we set $k = 1$).

## 3.2 WHY A SIMPLE HEAD SUFFICES AFTER R² MERGING

**Notation.** Let the penultimate features be $\phi(x;\theta) \in \mathbb{R}^d$. For each task $i \in [N]$ and label $y \in \{0,1\}$ (Fake=1, Real=0), we define the corresponding class means and covariances

$$\mu_{i,y}(\theta) \;:=\; \mathbb{E}[\phi(x;\theta) \mid i, y], \qquad \Sigma_{i,y}(\theta) \;:=\; \text{Cov}[\phi(x;\theta) \mid i, y]. \tag{7}$$

Then, the *Real–Fake* separation vector is defined as:

$$\Delta_i^{\text{RF}}(\theta) \;:=\; \mu_{i,1}(\theta) - \mu_{i,0}(\theta). \tag{8}$$

and the Jacobian at $\theta_0$ is written as:

$$J(x) \;:=\; \frac{\partial\phi(x;\theta)}{\partial\theta}\bigg|_{\theta=\theta_0} \in \mathbb{R}^{d\times D}, \qquad H_i \;:=\; \mathbb{E}[J(x) \mid i, 1] \;-\; \mathbb{E}[J(x) \mid i, 0]. \tag{9}$$

We use $\|\cdot\|_2$ for vector Euclidean norm, $\|\cdot\|_{\text{op}}$ for the matrix operator norm, and $\|\cdot\|_F$ for the Frobenius norm. For a unit vector $u$, let $P_{u^\perp} := I - uu^\top$ denote the orthogonal projector. We use $\angle(a,b)$ for the principal angle between nonzero vectors, defined by $\cos\angle(a,b) = \langle a,b\rangle/(\|a\|_2\|b\|_2)$. Unless otherwise specified, $v$ denotes the top right singular vector of the centered task matrix $M_c$ (§3.1).

**Theoretical Conditions (R1–R3).** We prove the following mild properties around $\theta_0$ (§A):

**(R1) Local linearity with bounded remainder.** For small $\Delta\theta$,

$$\Delta_i^{\text{RF}}(\theta_0 + \Delta\theta) \;=\; \Delta_i^{\text{RF}}(\theta_0) \;+\; H_i\,\Delta\theta \;+\; R_i(\Delta\theta), \qquad \|R_i(\Delta\theta)\| \leq C\,\|\Delta\theta\|^2. \tag{10}$$

*i.e., a small parameter displacement induces an approximately linear change in the Real–Fake separation through $H_i$.*

**(R2) Recovery of a shared Real axis by SVD.** Writing $\tau_i = a_i v^\star + \zeta_i$ with $\mathbb{E}[\zeta_i] = 0$ and bounded covariance, the top right singular vector $v$ of $M_c$ satisfies

$$\sin\angle(v, v^\star) \;\leq\; \gamma, \tag{11}$$

*i.e., SVD on $M_c$ recovers (up to a small angle $\gamma$) the common Real direction $v^\star$.*

**(R3) Off-axis control after truncation and norm matching.** With *layerwise top-$r$ truncation* of centered residuals and per-task norm matching,

$$\big\|P_{u^\perp} H_i\big(\eta\,\tau_{\text{merge}}^{\text{res}}\big)\big\| \;\leq\; \varepsilon', \big\|H_i\,\tau_{\text{core}}\big\| \qquad \text{for some } \varepsilon' \in [0,1). \tag{12}$$

*i.e., retaining only the leading singular components and equalizing residual norms bounds the off-axis response relative to the core push.*

**Proposition 1** (Directional alignment and averaged-head sufficiency under R²M). *Let $\theta_\star = \theta_0 + \tau_{\text{core}} + \eta\,\tau_{\text{merge}}^{\text{res}}$ be the R²M parameters from §3.1. Under* (R1)–(R3)*, there exists a unit $u \in \mathbb{R}^d$ such that the merged Real–Fake separation vectors are nearly collinear:*

$$\sin\angle\big(\Delta_i^{\text{RF}}(\theta_\star), u\big) \;\leq\; \frac{\varepsilon}{1-\varepsilon}, \qquad \forall i \in [N]. \tag{13}$$

We denote by $w_i^{\mathrm{sp}}$ the linear classification head (logit weights) of specialist $i$, and define their average head accordingly.

$$\bar{w} := \frac{1}{N} \sum_{i=1}^{N} w_i^{\mathrm{sp}}. \tag{14}$$

Then, there exist positive scalars $c_i > 0$ and a vector $q$ such that

$$w_i^{\mathrm{sp}} \approx c_i\, q, \qquad \bar{w} \approx \bar{c}\, q, \ \ \bar{c} = \frac{1}{N} \sum_i c_i > 0, \tag{15}$$

and the scores $s_i(x) = \langle w_i^{\mathrm{sp}}, \phi(x; \theta_\star) \rangle$ and $\bar{s}(x) = \langle \bar{w}, \phi(x; \theta_\star) \rangle$ differ only by a positive, task-dependent scaling. Consequently, the score rankings are preserved, and the AUC match up to this rescaling.

**Intuition and implications for deepfakes.** **(R2)** shows that applying SVD to centered task vectors recovers a shared Real axis; the core update moves every task along this axis. **(R3)** ensures that top-$r$ truncation and norm matching suppress off-axis drift from residuals, so that the Real–Fake separation vectors $\{\Delta_i^{\mathrm{RF}}(\theta_\star)\}_i$ concentrate in a narrow cone around a common direction $u$ in equation 13. This collinearity implies specialist heads align to the same effective direction on $\phi(x; \theta_\star)$; their average $\bar{w}$ remains aligned, making a *single, simple head* effective without loss in AUC. This behavior matches the structure of deepfake detection: Real cues are stable and shared, whereas Fake cues are generator-specific and volatile. The proposed $\mathrm{R}^2\mathrm{M}$ exploits this asymmetry by isolating a low-rank Real core and recombining denoised, balanced Fake residuals, yielding a deployment-friendly detector that retains in-domain strength and generalizes to unseen generators.

## 4 EXPERIMENTS

**Benchmark and protocols (DF40).** We evaluated on DF40 (Yan et al., 2024b), a recent and comprehensive deepfake detection benchmark that implements *40* distinct manipulation/generation methods across four *forgery categories*: face swapping (FS), face reenactment (FR), entire-face synthesis (EFS), and face editing (FE). We followed the DF40 naming of data "domains" (e.g., FF++ (Rossler et al., 2019) and Celeb-DF/CDF (Li et al., 2020c)). DF40 standardizes three evaluation protocols: (i) *Protocol 1 (cross-forgery, same domain)*: train and test within the same data domain while varying forgery methods; (ii) *Protocol 2 (cross-domain, same forgery)*: train and test on the same forgery category while changing the data domain; (iii) *Protocol 3 (unknown forgery & domain)*: train on seen forgeries/domains and test on *unseen* forgeries and domains to simulate open-set conditions. These protocol definitions, the four-category taxonomy (FS/FR/EFS/FE), and the FF++/CDF domains were adopted from DF40.

**Model roster and training policy.** We trained *three specialists*, one per forgery category: $\theta_{\mathrm{FS}}$ on the union of FS methods, $\theta_{\mathrm{FR}}$ on the union of FR methods, and $\theta_{\mathrm{EFS}}$ on the union of EFS methods (8 methods per category; total 24 methods). In addition, we trained a single *all-in-one* model jointly on the union of all these 24 methods. Across all models, the backbone is **CLIP-L/14** (Radford et al., 2021); we use its `pooler_output` as the embedding and a binary linear head trained with standard cross-entropy. Our goal is *not* to maximize in-domain AUC but to study model merging as a simple, training-free framework to cope with the rapidly proliferating forgery types. The specialist recipe is *model-agnostic*: any stronger detector can replace our specialists without changing the merge. All training strictly followed the official DF40 splits; no data from Protocol 3 was used for training or tuning. Merging was entirely training-free; hyperparameters ($k = 1, r, \alpha$) were chosen once on seen validation (Protocols 1–2) and reused across protocols. Hyperparameter ablations appear in § C.4, and the full list of forgery methods with the optimization schedule is provided in § C.

**Metrics: seen retention and unseen transfer.** On each task $i$, we evaluated the *image-level* area under the ROC curve (Fawcett, 2006), $\mathrm{AUC}_i(\theta) \in [0, 1]$, on the held-out test split $\mathcal{D}_i^{\mathrm{te}}$. We quantified matching between the merged model and the specialist on its own task via the per-task AUC drop : $\mathrm{Drop}_i := \mathrm{AUC}_i(\theta_i) - \mathrm{AUC}_i(\theta_\star)$, and aggregated by the worst-case: $\mathrm{Drop}_{\max} := \max_{i \in [N]} \mathrm{Drop}_i$. Smaller values are better ; we reported these empirically. Let $\mathcal{D}_{\mathrm{unseen}}$ denote data from generators or datasets not used for finetuning, and let $\mathrm{AUC}_{\mathrm{unseen}}(\theta)$ represent the corresponding AUC. We compared the merged model to specialist baselines via $\mathrm{Gain}_{\mathrm{unseen}} := \mathrm{AUC}_{\mathrm{unseen}}(\theta_\star) - \max_{i \in [N]} \mathrm{AUC}_{\mathrm{unseen}}(\theta_i)$, where positive values indicate improved zero-shot generalization over the best specialist.

Table 1: **Seen AUC (higher is better).** Columns are category×domain + domain-wise means. Best results are shown in **bold**, second-best are underlined.

| | Protocol1 - FF | | | | Protocol2 - CDF | | | |
|---|---|---|---|---|---|---|---|---|
| Method | FS | FR | EFS | *Mean* | FS | FR | EFS | *Mean* |
| *Our DF40-compliant training* | | | | | | | | |
| Specialist–FS | **0.995** | 0.912 | 0.766 | 0.891 | **0.959** | 0.690 | 0.603 | **0.751** |
| Specialist–FR | 0.923 | **0.999** | 0.742 | 0.888 | 0.505 | **0.915** | 0.099 | 0.506 |
| Specialist–EFS | 0.676 | 0.814 | **0.999** | 0.830 | 0.618 | 0.621 | **0.989** | 0.742 |
| All-in-one | 0.962 | 0.997 | 0.978 | **0.979** | 0.759 | 0.860 | 0.366 | 0.662 |
| *Training-free merging baselines* | | | | | | | | |
| Weight Averaging | 0.968 | **0.997** | 0.982 | 0.982 | 0.825 | 0.909 | 0.767 | 0.834 |
| Task Arithmetic | 0.956 | 0.995 | 0.965 | 0.972 | 0.741 | 0.888 | 0.926 | 0.852 |
| TIES-Merging | 0.959 | 0.993 | 0.961 | 0.971 | 0.819 | 0.898 | 0.891 | 0.869 |
| CART | 0.976 | 0.994 | 0.995 | **0.988** | 0.851 | 0.874 | 0.907 | 0.877 |
| **R$^2$M (ours)** | **0.977** | 0.992 | **0.996** | **0.988** | **0.902** | **0.912** | **0.942** | **0.919** |

Table 2: **Unseen AUC (higher is better)** on DF40 Protocol 3. *Unseen datasets:* DeepFaceLab (Liu et al., 2023b), HeyGen (hey), Midjourney (mid), WhichIsReal (whi), StarGAN (Choi et al., 2018), StarGAN v2 (Choi et al., 2020), StyleCLIP (Patashnik et al., 2021), CollabDiff (Huang et al., 2023).

| Method | DeepFace Lab | Hey Gen | Mid journey | WhichIs Real | Star Gan | StarGan v2 | Style Clip | Collab Diff | *Mean* |
|---|---|---|---|---|---|---|---|---|---|
| *Our DF40-compliant training* | | | | | | | | | |
| Specialist–FS | **0.892** | **0.573** | 0.334 | 0.364 | 0.887 | 0.640 | 0.608 | 0.596 | 0.612 |
| Specialist–FR | 0.812 | 0.546 | **0.603** | 0.228 | 0.697 | 0.433 | 0.053 | 0.141 | 0.439 |
| Specialist–EFS | 0.709 | 0.398 | 0.561 | **0.694** | **0.901** | **0.777** | **0.952** | **0.997** | **0.749** |
| All-in-one | 0.884 | 0.561 | 0.177 | 0.512 | 0.860 | 0.714 | 0.761 | 0.854 | 0.665 |
| *Training-free merging baselines (same backbone)* | | | | | | | | | |
| WA | 0.930 | **0.626** | 0.613 | 0.455 | 0.953 | 0.728 | 0.635 | 0.848 | 0.724 |
| TA | 0.887 | 0.541 | **0.695** | 0.255 | 0.891 | 0.579 | 0.504 | 0.622 | 0.622 |
| TM | 0.897 | 0.579 | 0.545 | 0.327 | 0.912 | 0.598 | 0.668 | 0.627 | 0.644 |
| CART | 0.943 | 0.608 | 0.559 | **0.497** | **0.975** | **0.780** | 0.813 | 0.955 | 0.766 |
| **R$^2$M (ours)** | **0.946** | 0.617 | 0.551 | 0.492 | 0.973 | 0.778 | **0.860** | **0.974** | **0.774** |

For **Protocols 1–2** (seen settings), we evaluated *Seen-task retention* using the DROP metric (lower is better). For **Protocol 3** (unseen setting), we summarized zero-shot generalization by *Unseen transfer* (GAIN) (higher is better). Hyperparameters and thresholds were selected on seen validation and were not tuned on Protocol 3. We followed DF40's official train/test splits and per-protocol settings; no training was performed during merging. We evaluated standard merging variants alongside our method; precise formulations and hyperparameter grids appear in §B.1.

**Tuning protocol.** For parameters that change the *internal composition* of task vectors (TIES (Yadav et al., 2023) sparsity, CART(Choi et al., 2024a) rank, R$^2$M rank), we sweep $\{0.1, 0.3, 0.5, 0.7\}$. For *global magnitude* scalars (Task Arithmetic (Ilharco et al., 2022), CART), we sweep $\{0.5, 1.0\}$. For R$^2$M, we sweep the normalization coefficient $\alpha \in \{0.4, 0.5, 0.6\}$ in $\eta_{\text{eff}} = \alpha \|\tau_{\text{core}}\| / \|\tau_{merge}^{res}\|$ in equation 6. To ensure fair comparison, we used the same grids across all merged models, together with the same averaged head.

### 4.1 SEEN RETENTION ON DF40 (PROTOCOLS 1–2)

We first summarized per-category AUCs on the seen domains (FF++; Protocol 1) in Table 1.

*(i) Training-free model merging fits deepfake detection.* Even plain *Weight Averaging* achieves a strong FF mean AUC of **0.982**, essentially matching the jointly trained *All-in-one* model (0.979). This indicates substantial cross-task parameter sharing in this domain and suggests that weight-space merging is a natural, effective paradigm for deepfake detection.

*(ii) CART and R$^2$M retain specialists almost perfectly on FF (saturation).* Both *CART* and *R$^2$M* reach the same FF mean AUC **0.988**, with category-wise values tightly tracking the specialists (e.g., FS: 0.976/0.977 vs 0.995; FR: 0.994/0.992 vs 0.999; EFS: 0.995/0.996 vs 0.999). In terms of retention, this corresponds to very small DROP (on the order of $10^{-2}$), showing that on the in-domain FF setting the problem is near-saturated across merging methods.

*(iii) Cross-domain seen transfer (CDF; Protocol 2).* When the domain shifts from FF++ to Celeb-DF while keeping the manipulation types fixed, our method shows a clear advantage over other training-free mergers. R$^2$M attains the best CDF mean AUC (**0.919**), outperforming CART (0.877), TIES (0.869), Task Arithmetic (0.852), and Weight Averaging (0.834). Although each specialist remains the upper bound on its own category within CDF, the merged R$^2$M backbone generalizes substantially better than other closed-form mergers under this domain shift. This indicates that the Real-

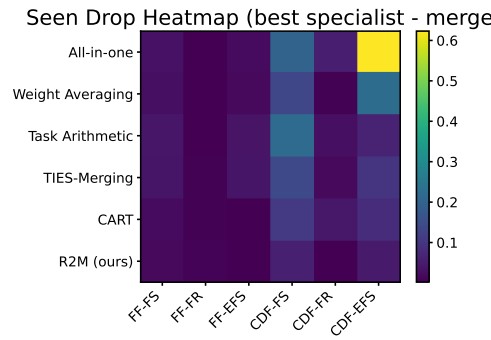 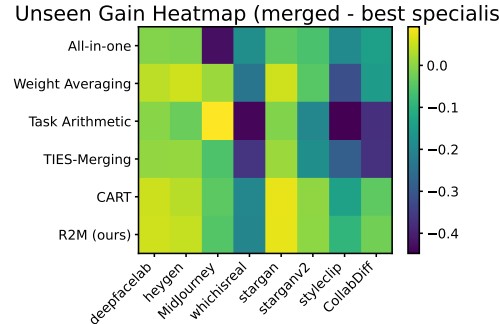

(a) **Seen-task retention (FF&CDF):** DROP = Best specialist AUC − Merged AUC (lower is better). Columns are {FF, CDF}×{FS, FR, EFS}.

(b) **Unseen transfer (Protocol 3):** GAIN = Merged AUC − Best specialist AUC (higher is better) over 8 unseen forgeries.

Figure 4: **Training-free merging summary (DF40).** Left: seen-task retention; smaller DROP is better. Right: unseen transfer; larger GAIN is better.

aware residual decomposition not only preserves seen-task skill on FF but also confers improved robustness across domains.

As summarized in Table 1, we quantify seen-task retention with DROP (best specialist AUC minus merged model AUC) in Fig.4a. Consistent with our three observations, $\mathbf{R^2M}$ exhibits the darkest cells (lower is better), while the *All-in-one* model shows notably lighter cells especially at **CDF–EFS**, indicating poor cross-domain retention. Closed-form mergers (WA/TIES/CART) are generally darker than *All-in-one*, but $\mathbf{R^2M}$ is uniformly strongest.

### 4.2 UNSEEN GENERALIZATION ON DF40 (PROTOCOL 3)

Table 2 reports per-forgery AUC on *unseen* generators, along with the macro mean; we contextualize results using GAIN in Fig.4b, defined as the improvement over the best specialist in each column.

*(i) Merging prevents catastrophic failures and improves macro performance.* While the *All-in-one* model collapses on `MidJourney` (AUC = **0.177**), all training-free merging variants avoid such failure (*e.g.*, Weight Averaging 0.613, CART 0.559, $R^2M$ 0.551). In macro terms, merging is consistently competitive or superior to All-in-one: *Weight Averaging* averages **0.724**, *CART* **0.766**, and $R^2M$ **0.774**, the best among training-free methods.

*(ii) Specialists reveal structure in the unseen space. Specialist–EFS* is notably strong on several unseen generators driven by image synthesis models, *e.g.*, `stargan` 0.901, `starganv2` 0.777, `styleclip` 0.952, `CollabDiff` 0.997, suggesting that entire-face synthesis induces artifacts aligned with EFS-trained cues. Conversely, for unfamiliar content-generation platforms such as `MidJourney`, specialist performance is limited (best specialist 0.603), and joint training can be brittle (All-in-one 0.177). In unseen transfer (Protocol 3; Fig.4b), $\mathbf{R^2M}$ shows the largest bright area GAIN, consistently outperforming the best specialist on the eight unseen forgeries, while other mergers exhibit mixed or negligible gains.

*(iii) $R^2M$ delivers the strongest overall zero-shot transfer, with balanced gains.* Relative to the best specialist per column, $R^2M$ shows positive GAIN on several unseen generators (`deepfacelab` +0.054, `heygen` +0.044, `stargan` +0.072, `starganv2` +0.001), while remaining competitive on the rest (`MidJourney` −0.052, `whichisreal` −0.202, `styleclip` −0.092, `CollabDiff` −0.023). Importantly, the merged backbones (CART/$R^2M$) substantially reduce worst-case behavior compared to All-in-one, indicating that training-free merging yields more robust out-of-distribution generalization. *At the macro level*, both CART and $R^2M$ also outperform specialist-only baselines: their mean unseen AUCs

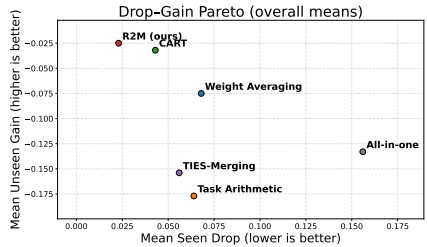

Figure 5: **Drop-Gain Pareto (overall means).** Each point is a merged model; $x$=mean DROP (lower is better), $y$=mean GAIN (higher is better).

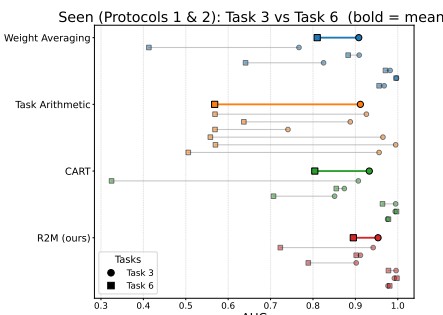

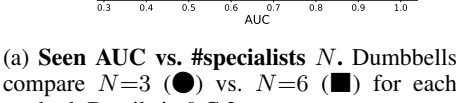

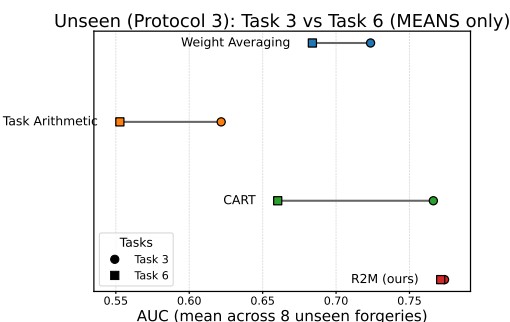

(a) **Seen AUC vs. #specialists** $N$**.** Dumbbells compare $N{=}3$ (●) vs. $N{=}6$ (■) for each method; Details in § C.3
.

(b) **Unseen AUC vs. #specialists** $N$**.** Each dumbbell shows the method's mean AUC over the eight unseen forgeries, comparing $N{=}3$ (●) to $N{=}6$ (■).

Figure 6: **Scaling with #specialists** $N$ **(Task 3 $\rightarrow$ Task 6).** (a) Seen performance per category (means in bold); (b) Unseen performance as the mean over the eight forgeries. Across $N$, baselines (WA/TA/CART) shift more, while $\mathbf{R^2M}$ changes less, indicating stronger retention and more stable transfer when scaling up specialists.

(**0.766** and **0.774**, respectively) exceed the average of specialists (0.600) by large margins and even surpass the strongest single specialist (EFS, 0.749) by +0.017 (CART) and +0.025 ($R^2$M).

In the Drop–Gain Pareto plot (Fig.5), the *All-in-one* model sits far to the right, indicating the largest retention loss, whereas training-free mergers cluster on the left with markedly smaller DROP. Among these, $\mathbf{R^2M}$ lies in the upper-left corner, achieving the best results (lowest DROP, highest GAIN). *CART* comes next with similarly low DROP but a smaller GAIN. The remaining mergers are ordered mainly by GAIN: *Weight Averaging > TIES > Task Arithmetic*.

### 4.3 SCALABILITY AND INCREMENTAL MERGING

**Scaling with #specialists** $N$**.** We investigated how merging performance scales as the number of specialists grows from $N{=}3$ to $N{=}6$. The left panel of Fig.6 shows *seen* performance (AUC) per task/domain, while the right panel shows *unseen* performance (AUC) averaged over the eight forgeries. Across baselines, increasing $N$ causes noticeably larger shifts (longer dumbbells), reflecting greater degradation on seen tasks and more volatile transfer on unseen data. In contrast, $\mathbf{R^2M}$ exhibits consistently *smaller* movement, indicating better retention under scale.

**Fast integration of new deepfake methods.** We next study how $R^2$M behaves when a new forgery method appears after deployment. In practice, there are two distinct scenarios: (i) the new generator is semantically and visually close to an existing forgery family, and (ii) it is substantially different and lies far outside the training distribution.

*Case 1*: new generator similar to an existing family (Uniface). We first consider Uniface, an FS-style method that is not used when training the original specialists. Starting from three specialists trained on eight FS/FR/EFS generators (FS, FR, EFS), the merged model (FS+FR+EFS) already achieves strong performance on Uniface, with AUCs [0.977, 0.992, 0.996, 0.972] on (FS, FR, EFS, Uniface), respectively. We then train a dedicated Uniface specialist and merge four specialists (FS, FR, EFS, Uniface) with $R^2$M. After merging, the AUCs become [0.976, 0.991, 0.996, 0.989] on the same four tasks. Thus, even without an additional special-

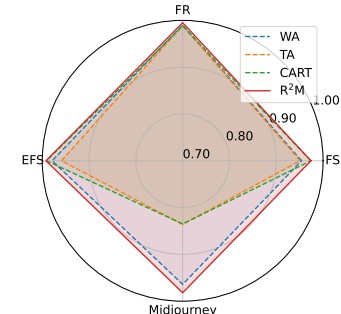

Figure 7: **Fast integration (AUC).** Radar plot over four forgery types (FS-FF, FR-FF, EFS-FF, Midjourney) comparing WA, TA, CART, and $R^2$M. Final AUCs: **WA** = [0.954, 0.990, 0.979, 0.964], **TA** = [0.951, 0.989, 0.959, 0.836], **CART** = [0.974, 0.988, 0.991, 0.835], $\mathbf{R^2M}$ = [0.974, 0.995, 0.992, 0.982]. $R^2$M delivers the highest Midjourney AUC (0.982) with no regression on legacy forgery types, demonstrating plasticity without forgetting.

ist, $R^2$M generalizes well to a new method that is aligned with an existing family (Uniface AUC 0.972), and once a specialist is available it further improves Uniface (0.989) while keeping the legacy FS/FR/EFS performance essentially unchanged (differences $\leq$ 0.001). This illustrates that $R^2$M can integrate "more of the same" forgery types with *minimal interference* to existing domains.

*Case 2*: new generator far from existing families (Midjourney). We then evaluate a previously unseen text-to-image generator (Midjourney), which is much more visually diverse than DF40's face-centric forgeries. We finetune a single specialist on DF40–Midjourney using an 80/20 train/test split with strict disjointness, and then update the deployed model via one-shot merging. Fig.7 summarizes four *forgery types* (FS-FF, FR-FF, EFS-FF, Midjourney) across four merging algorithms. The **R$^2$M** curve (red) encloses the largest area, indicating consistently strong detection across all forgery types, including the newly introduced Midjourney. **WA** forms the next largest envelope but remains slightly inside R$^2$M on most axes. In contrast, **TA** and **CART** exhibit notably smaller coverage on the Midjourney axis, reflecting limited plasticity to the new generator. Overall, these patterns suggest that WA serves as a competitive baseline, while the *per-task norm matching* in $R^2$M further stabilizes merging, integrating heterogeneous, newly arriving specialists *without* degrading legacy performance.

## 4.4 Merging specialists with heterogeneous real domains

Our analysis so far has used DF40, in which all real images are drawn from FF. One may therefore question whether the observed "shared Real" structure persists when the real distribution is more heterogeneous. We address this by constructing a more challenging setting where specialists are trained on *two* distinct real datasets and then merged.

**Setup.** In addition to the three FF-based specialists (FS-ff, FR-ff, EFS-ff), we train three more specialists that use **CDF** as the real source (FS-cdf, FR-cdf, EFS-cdf) while keeping the same forgery families. We then merge all six specialists into a single detector using either CART or $R^2$M, and evaluate AUC on each of

Table 3: **Merging six specialists with heterogeneous real distributions.** AUC on six domains for the individual specialists (FS-ff, FR-ff, EFS-ff, FS-cdf, FR-cdf, EFS-cdf), and for the merged models obtained by CART and $R^2$M.

| Model | FS-ff | FR-ff | EFS-ff | FS-cdf | FR-cdf | EFS-cdf |
|---|---|---|---|---|---|---|
| FS-ff | 0.995 | 0.912 | 0.766 | 0.959 | 0.696 | 0.621 |
| FR-ff | 0.923 | 0.9996 | 0.742 | 0.503 | 0.923 | 0.106 |
| EFS-ff | 0.676 | 0.814 | 0.999 | 0.635 | 0.608 | 0.988 |
| FS-cdf | 0.754 | 0.647 | 0.616 | 0.9999 | 0.989 | 0.982 |
| FR-cdf | 0.557 | 0.849 | 0.532 | 0.991 | 0.9999 | 0.929 |
| EFS-cdf | 0.329 | 0.244 | 0.923 | 0.955 | 0.961 | 0.9999 |
| CART | 0.867 | 0.957 | 0.954 | 0.998 | 0.994 | 0.996 |
| $R^2$M | 0.901 | 0.969 | 0.962 | 0.996 | 0.994 | 0.996 |

the six domains ({FS-ff, FR-ff, EFS-ff, FS-cdf, FR-cdf, EFS-cdf}). Training details and the full construction of this setting are provided in the § B.2

**Results.** Table 3 reports the cross-domain performance of six individual specialists, with the merged models obtained by CART and $R^2$M. As expected, asking a single model to cover six specialists trained on two different real distributions is substantially more challenging than the original FF++-only case. Nevertheless, $R^2$M consistently achieves higher AUC than CART on the harder FF++ domains, while matching CART on the saturated CDF domains. This indicates that the SVD-based Real core in $R^2$M continues to provide a stabilizing shared subspace even when Real comes from multiple datasets, and that $R^2$M *degrades more gracefully* than CART as real diversity increases.

## 5 Conclusion

We framed deepfake detection as a structurally natural case for model merging: specialists share a binary decision while differing in generator-specific artifacts. Our probes showed that simple weight averaging preserves Real structure and suppresses generator-specific cues, motivating a domain-tailored merge. We introduced **R$^2$M**, a training-free parameter-space procedure that isolates a shared Real component and aggregates denoised, norm-matched Fake residuals. The resulting single detector retains in-domain strength, improves transfer to unseen generators, and is composable: new forgeries are handled by fine-tuning one specialist and merging it with the existing model, without retraining prior components. Looking ahead, scaling to broader specialist collections covering additional generators, backbones, and modalities should further amplify the benefits of merging. Beyond this, we see promising directions in extending hierarchical, self-expanding merging schemes that group related specialists and recursively compose their cores. Another important avenue is robustness checks against poisoned or untrusted specialists, moving toward a safe, continuously updatable deepfake defense pipeline.

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

# A  THEORY PROOFS

## A.1  PRELIMINARIES: ASSUMPTIONS, NOTATION, AND TOOLS

We gather the standing assumptions and basic results used in the proofs.

**Assumptions (Smoothness).**  Throughout, the feature map $\phi(\cdot; \theta)$ is twice continuously differentiable in $\theta$ on a neighborhood $\mathcal{N}$ of $\theta_0$. Its Jacobian

$$J(x; \theta) := \frac{\partial \phi(x; \theta)}{\partial \theta} \in \mathbb{R}^{d \times D} \tag{A.1}$$

is locally $L$-Lipschitz in $\theta$ uniformly in $x \in \mathcal{X}$:

$$\|J(x; \theta_1) - J(x; \theta_2)\|_{\mathrm{op}} \le L \|\theta_1 - \theta_2\|_2 \qquad \forall \theta_1, \theta_2 \in \mathcal{N}. \tag{A.2}$$

**Notation (recap).**  We restate only the symbols used in the proofs for self-containment. For task $i \in [N]$ and label $y \in \{0, 1\}$ (Fake=1, Real=0), recall

$$\mu_{i,y}(\theta) = \mathbb{E}[\phi(x; \theta) \mid i, y], \quad \Sigma_{i,y}(\theta) = \mathrm{Cov}[\phi(x; \theta) \mid i, y], \quad \Delta_i^{\mathrm{RF}}(\theta) = \mu_{i,1}(\theta) - \mu_{i,0}(\theta). \tag{A.3}$$

Write $H_i := \mathbb{E}[J(x; \theta_0) \mid i, 1] - \mathbb{E}[J(x; \theta_0) \mid i, 0]$ and use $\| \cdot \|_2$ (vector), $\| \cdot \|_{\mathrm{op}}$ (operator), and $\| \cdot \|_F$ (Frobenius) norms. For a unit $u \in \mathbb{R}^d$, $P_{u^\perp} := I - uu^\top$ is the orthogonal projector. The angle between nonzero vectors satisfies $\cos \angle(a, b) = \langle a, b \rangle / (\|a\|_2 \|b\|_2)$.

**Task vectors and SVD.**  Let $\tau_i := \theta_i - \theta_0$, $\bar{\tau} := \frac{1}{N} \sum_{i=1}^N \tau_i$, and

$$M = \begin{bmatrix} \tau_1^\top \\ \vdots \\ \tau_N^\top \end{bmatrix} \in \mathbb{R}^{N \times D}, \qquad M_c = M - \mathbf{1} \bar{\tau}^\top. \tag{A.4}$$

Let $M_c = U_c \Sigma_c V_c^\top$ be a compact SVD and denote by $v$ the top right singular vector.

**Residuals and layerwise top-$r$ truncation.**  Define the mean-centered residuals $\delta_i := \tau_i - \bar{\tau}$. For any layer $\ell$ and task $i$, let $W_i^{(\ell)}$ be the matrix slice extracted from $\delta_i$ (attention/MLP blocks). Write its compact SVD $W_i^{(\ell)} = U_i^{(\ell)} \Sigma_i^{(\ell)} V_i^{(\ell)\top}$ and keep only the top-$r$ singular components:

$$\tilde{W}_i^{(\ell)} := \mathrm{SVD}_r(W_i^{(\ell)}) = U_{i,r}^{(\ell)} \Sigma_{i,r}^{(\ell)} (V_{i,r}^{(\ell)})^\top. \tag{A.5}$$

Replacing every targeted layer $W_i^{(\ell)}$ by $\tilde{W}_i^{(\ell)}$ and reassembling yields the *truncated residual* $\tilde{\delta}_i$. We use $\mathrm{SVD}_r(\cdot)$ throughout to denote the operator that retains the top-$r$ singular components at the layer level (cf. Sec. 3.1).

**Matrix tools: perturbation and low-rank approximation.**  We rely on classical subspace perturbation bounds (Wedin's and Davis–Kahan's sin–$\Theta$ theorems) and on the Eckart–Young–Mirsky theorem for best low-rank approximation (Wedin, 1972; Davis & Kahan, 1970; Eckart & Young, 1936). These are standard in PCA/spectral analyses and low-rank denoising (Von Luxburg, 2007), and we invoke them in R2 (subspace recovery) and R3 (tail-energy control), respectively.

## A.2  PROOF OF R1 (LOCAL LINEARITY WITH BOUNDED REMAINDER)

**Lemma 1** (R1).  *For small $\Delta\theta$,*

$$\Delta_i^{\mathrm{RF}}(\theta_0 + \Delta\theta) = \Delta_i^{\mathrm{RF}}(\theta_0) + H_i \Delta\theta + R_i(\Delta\theta), \qquad \|R_i(\Delta\theta)\|_2 \le C \|\Delta\theta\|_2^2, \tag{A.6}$$

*for some constant $C > 0$ depending on the Lipschitz constant in equation A.2.*

*Proof.* Fix $i$ and $y \in \{0, 1\}$. By the mean-value form of Taylor's theorem for vector-valued maps,

$$\mu_{i,y}(\theta_0 + \Delta\theta) = \mu_{i,y}(\theta_0) + \left( \mathbb{E}[J(x; \theta_0) \mid i, y] \right) \Delta\theta + r_{i,y}(\Delta\theta), \tag{A.7}$$

with remainder bounded (using equation A.2 and Jensen) by

$$\|r_{i,y}(\Delta\theta)\|_2 \; \leq \; \tfrac{L}{2} \, \|\Delta\theta\|_2^2. \tag{A.8}$$

Subtracting the expressions for $y = 0$ from $y = 1$ yields

$$\Delta_i^{\mathrm{RF}}(\theta_0 + \Delta\theta) \; = \; \Delta_i^{\mathrm{RF}}(\theta_0) \; + \; H_i \, \Delta\theta \; + \; R_i(\Delta\theta), \qquad R_i(\Delta\theta) := r_{i,1}(\Delta\theta) - r_{i,0}(\Delta\theta), \tag{A.9}$$

and thus $\|R_i(\Delta\theta)\|_2 \leq L \, \|\Delta\theta\|_2^2 =: C\|\Delta\theta\|_2^2.$ □

### A.3 Proof of R2 (SVD recovers the shared Real axis)

**Lemma 2** (R2). *Assume the decomposition $\tau_i = a_i v^\star + \zeta_i$ with $\mathbb{E}[\zeta_i] = 0$, $\mathrm{Cov}(\zeta_i) \preceq \sigma^2 I$, and $\mathrm{Var}(a_i) = \sigma_a^2 > 0$. Let $M_c$ be the row-centered task matrix and $v$ its top right singular vector. If the spectral gap $\sigma_a\|v^\star\|_2 \gg \sigma$ holds, then*

$$\sin\angle(v,\, v^\star) \; \leq \; \gamma \; := \; \frac{\|Z_c^\top\|_{\mathrm{op}}}{\|A_c v^\star\|_2 - \|Z_c^\top\|_{\mathrm{op}}}\,, \tag{A.10}$$

*where $A_c$ stacks $(a_i - \bar{a})$ and $Z_c$ stacks $(\zeta_i - \bar{\zeta})$.*

*Proof.* Centering removes the mean: $M_c = A_c v^{\star\top} + Z_c$, with $A_c \in \mathbb{R}^{N\times 1}$ (the column of $(a_i - \bar{a})$) and $Z_c \in \mathbb{R}^{N\times D}$. Then

$$M_c^\top M_c \; = \; v^\star A_c^\top A_c v^{\star\top} \; + \; v^\star A_c^\top Z_c \; + \; Z_c^\top A_c v^{\star\top} \; + \; Z_c^\top Z_c. \tag{A.11}$$

The rank-one signal part is $S := v^\star A_c^\top A_c v^{\star\top}$ with top eigenvector $v^\star$ and eigenvalue $\|A_c v^\star\|_2^2 = \|A_c\|_2^2 \, \|v^\star\|_2^2$. The remainder $E := M_c^\top M_c - S$ satisfies $\|E\|_{\mathrm{op}} \leq 2\|Z_c^\top\|_{\mathrm{op}}\|A_c\|_2 + \|Z_c^\top Z_c\|_{\mathrm{op}}$; by the covariance bound and concentration (or in expectation), we have $\|Z_c^\top\|_{\mathrm{op}} = O(\sigma\sqrt{N})$ and $\|A_c\|_2 = \Theta(\sigma_a\sqrt{N})$. Under the stated gap condition, the Davis–Kahan sin–$\Theta$ theorem gives

$$\sin\angle(v,\, v^\star) \; \leq \; \frac{\|E\|_{\mathrm{op}}}{\lambda_1(S) - \lambda_2(S) - \|E\|_{\mathrm{op}}} \; \leq \; \frac{\|Z_c^\top\|_{\mathrm{op}}}{\|A_c v^\star\|_2 - \|Z_c^\top\|_{\mathrm{op}}}, \tag{A.12}$$

where we used that $S$ is rank-one (so $\lambda_2(S) = 0$) and absorbed constants. The bound vanishes as $\sigma/(\sigma_a\|v^\star\|_2) \to 0$. □

### A.4 Proof of R3 (Off-axis control after top-$r$ truncation and norm matching)

**Lemma 3** (R3). *Let $\delta_i := \tau_i - \bar{\tau}$ and let $\tilde{\delta}_i$ be their layerwise top-$r$ truncated versions (best rank-$r$ approximations in $\|\cdot\|_F$). Define norm-matched residuals*

$$m_{\mathrm{mean}} := \frac{1}{N}\sum_{j=1}^N \|\tilde{\delta}_j\|_2, \qquad \hat{\delta}_i := m_{\mathrm{mean}}\frac{\tilde{\delta}_i}{\|\tilde{\delta}_i\|_2 + \varepsilon}, \tag{A.13}$$

*for a small $\varepsilon > 0$. Let $\tau_{\mathrm{res}}^{\mathrm{merge}} := \frac{1}{N}\sum_{j=1}^N \hat{\delta}_j$ and $\theta_\star = \theta_0 + \tau_{\mathrm{core}} + \eta\,\tau_{\mathrm{res}}^{\mathrm{merge}}$. For any unit $u \in \mathbb{R}^d$,*

$$\big\|P_{u^\perp} H_i(\eta\,\tau_{\mathrm{res}}^{\mathrm{merge}})\big\| \; \leq \; \varepsilon'\,\big\|H_i\,\tau_{\mathrm{core}}\big\| \qquad \text{for some } \varepsilon' \in [0,1), \tag{A.14}$$

*where one can take*

$$\varepsilon' \; = \; C\,\alpha\,(\kappa_u + \bar{\tau}_r)\,\frac{\|H_i\|_{\mathrm{op}}\,m_{\mathrm{mean}}}{\|H_i\tau_{\mathrm{core}}\|_2}, \qquad \kappa_u := \frac{1}{N}\sum_{j=1}^N \frac{\|P_{u^\perp}\delta_j\|_F}{\|\delta_j\|_F}, \quad \bar{\tau}_r := \frac{1}{N}\sum_{j=1}^N \frac{\big(\sum_{\ell>r}\sigma_\ell(\delta_j)^2\big)^{1/2}}{\|\delta_j\|_F}, \tag{A.15}$$

*for a universal constant $C > 0$. In particular, fixing $r$ and choosing $\eta = \alpha\|\tau_{\mathrm{core}}\|_2$ with $\alpha > 0$ small enough ensures $\varepsilon' < 1$.*

*Proof.* By submultiplicativity and orthogonality of $P_{u^\perp}$,

$$\big\|P_{u^\perp} H_i(\eta\,\tau_{\mathrm{res}}^{\mathrm{merge}})\big\| \; \leq \; \|H_i\|_{\mathrm{op}}\,\eta\,\big\|P_{u^\perp}\tau_{\mathrm{res}}^{\mathrm{merge}}\big\| \; \leq \; \|H_i\|_{\mathrm{op}}\,\eta\,\frac{1}{N}\sum_{j=1}^N \big\|P_{u^\perp}\hat{\delta}_j\big\|. \tag{A.16}$$

For each task $j$, by the triangle inequality and the Eckart–Young–Mirsky theorem,

$$\|P_{u^\perp}\tilde{\delta}_j\|_F \;\leq\; \|P_{u^\perp}\delta_j\|_F + \|\delta_j - \tilde{\delta}_j\|_F \;\leq\; \|P_{u^\perp}\delta_j\|_F + \left(\sum_{\ell>r}\sigma_\ell(\delta_j)^2\right)^{1/2}. \qquad (A.17)$$

Aggregating layerwise slices to the parameter vector and using norm matching $\|\hat{\delta}_j\|_2 \approx m_{\text{mean}}$ (up to a universal constant $C$ due to block aggregation) yields

$$\|P_{u^\perp}\hat{\delta}_j\|_2 \;\leq\; C\, m_{\text{mean}} \left(\frac{\|P_{u^\perp}\delta_j\|_F}{\|\delta_j\|_F} + \frac{\left(\sum_{\ell>r}\sigma_\ell(\delta_j)^2\right)^{1/2}}{\|\delta_j\|_F}\right). \qquad (A.18)$$

Averaging over $j$ gives

$$\frac{1}{N}\sum_{j=1}^N \|P_{u^\perp}\hat{\delta}_j\|_2 \;\leq\; C\, m_{\text{mean}}\,(\kappa_u + \bar{\tau}_r). \qquad (A.19)$$

Plugging this bound into equation A.16 and setting $\eta = \alpha\|\tau_{\text{core}}\|_2$,

$$\left\|P_{u^\perp}H_i(\eta\,\tau_{\text{res}}^{\text{merge}})\right\| \;\leq\; C\,\|H_i\|_{\text{op}}\,\alpha\,(\kappa_u + \bar{\tau}_r)\,m_{\text{mean}}\,\|\tau_{\text{core}}\|_2. \qquad (A.20)$$

Since $\|H_i\tau_{\text{core}}\|_2 \geq c_0\,\|\tau_{\text{core}}\|_2$ for some task-dependent $c_0 > 0$ (nondegenerate response along the core), we obtain

$$\left\|P_{u^\perp}H_i(\eta\,\tau_{\text{res}}^{\text{merge}})\right\| \;\leq\; \underbrace{C\,\alpha\,(\kappa_u + \bar{\tau}_r)\,\frac{\|H_i\|_{\text{op}}\,m_{\text{mean}}}{\|H_i\tau_{\text{core}}\|_2}}_{=:\,\varepsilon'}\,\|H_i\tau_{\text{core}}\|_2. \qquad (A.21)$$

Choosing $\alpha > 0$ small enough ensures $\varepsilon' < 1$, completing the proof. $\qquad\square$

## A.5 PROOF OF PROPOSITION 1

**Proposition 2** (Restatement of Proposition 1). *Let $\theta_\star = \theta_0 + \tau_{\text{core}} + \eta\,\tau_{\text{merge}}^{\text{res}}$. Under (R1)–(R3), there exists a unit $u \in \mathbb{R}^d$ such that*

$$\sin\angle\big(\Delta_i^{\text{RF}}(\theta_\star),\, u\big) \;\leq\; \frac{\varepsilon}{1-\varepsilon}\,, \qquad \forall i \in [N]. \qquad (A.22)$$

*Let $w_i^{\text{sp}}$ be the linear head (logit weights) of specialist $i$, and $\bar{w} = \frac{1}{N}\sum_i w_i^{\text{sp}}$. Then there exist $c_i > 0$ and a vector $q$ with*

$$w_i^{\text{sp}} \approx c_i\,q, \qquad \bar{w} \approx \bar{c}\,q, \;\; \bar{c} = \frac{1}{N}\sum_i c_i > 0, \qquad (A.23)$$

*and the scores $s_i(x) = \langle w_i^{\text{sp}}, \phi(x;\theta_\star)\rangle$ and $\bar{s}(x) = \langle \bar{w}, \phi(x;\theta_\star)\rangle$ differ by a positive scale, hence preserve AUC.*

*Proof.* By R2, the top right singular vector $v$ of $M_c$ approximates the shared Real axis $v^\star$: $\sin\angle(v, v^\star) \leq \gamma$. Consider the R$^2$M update $\Delta\theta = \tau_{\text{core}} + \eta\,\tau_{\text{res}}^{\text{merge}}$. By R1 (local linearity),

$$\Delta_i^{\text{RF}}(\theta_\star) = \Delta_i^{\text{RF}}(\theta_0) + H_i\tau_{\text{core}} + H_i(\eta\,\tau_{\text{res}}^{\text{merge}}) + R_i(\Delta\theta), \quad \|R_i(\Delta\theta)\| \leq C\|\Delta\theta\|_2^2. \quad (A.24)$$

Let $u$ be the (unit) common response direction to the core, i.e. $H_i v \approx s_i u$ with $s_i > 0$ (R2 gives dominance of this mode; continuity yields a uniform $u$). Then $H_i\tau_{\text{core}} = (v^\top\tau_{\text{core}})H_i v \approx (v^\top\tau_{\text{core}})s_i u$. By R3 (off-axis control after truncation and norm matching),

$$\left\|P_{u^\perp}H_i(\eta\,\tau_{\text{res}}^{\text{merge}})\right\|_2 \;\leq\; \varepsilon'\left\|H_i\tau_{\text{core}}\right\|_2, \qquad \varepsilon' \in [0,1). \qquad (A.25)$$

Absorbing $\|R_i(\Delta\theta)\| = O(\|\Delta\theta\|_2^2)$ into $\varepsilon$ (small $\eta$), the off-axis component of $\Delta_i^{\text{RF}}(\theta_\star)$ is at most an $\varepsilon$-fraction of the on-axis magnitude, giving

$$\sin\angle\big(\Delta_i^{\text{RF}}(\theta_\star),\, u\big) \;\leq\; \frac{\varepsilon}{1-\varepsilon}. \qquad (A.26)$$

Thus $\{\Delta_i^{\text{RF}}(\theta_\star)\}_i$ are nearly colinear (directional alignment).

For heads, denote the pooled within-class covariance by $\Sigma(\theta_\star)$. Since $\Delta_i^{\text{RF}}(\theta_\star) \approx \alpha_i u$ with $\alpha_i > 0$, any specialist head trained to separate Real/Fake aligns to the same effective direction $q$ on $\phi(x;\theta_\star)$ up to a positive scale $c_i$ (e.g., in the LDA idealization $w_i^\star \propto \Sigma(\theta_\star)^{-1}\Delta_i^{\text{RF}}(\theta_\star) = c_i q$). Hence $w_i^{\text{sp}} \approx c_i q$ and $\bar{w} \approx \bar{c}q$. Consequently $s_i(x)$ and $\bar{s}(x)$ differ by a positive scalar across inputs, preserving score rankings and AUC. $\qquad\square$

## B    IMPLEMENTATION DETAILS AND MERGING BASELINES

### B.1    BACKBONES, HEADS, AND CLOSED-FORM MERGING

Let $\theta_0$ be the pretrained weights and $\tau_i = \theta_i - \theta_0$ the task vector of specialist $i$ among $N$.

- **Pretrained ($\theta_0$):** zero-shot without finetuning.
- **Specialists ($\{\theta_i\}$):** one model per forgery method.
- **All-in-one:** single model trained on the union of seen forgeries.
- **Model-merging (closed-form, no retraining).** For any merged backbone $\hat{\theta}$, we attached the same averaged specialist's head $\bar{\phi} = \frac{1}{N}\sum_{i=1}^{N}\phi_i$ and evaluated $(\hat{\theta}, \bar{\phi})$ for all variants.
  - *Weight Averaging:* $\theta_{\mathrm{avg}} = \theta_0 + \frac{1}{N}\sum_{i=1}^{N}\tau_i$ (no hyperparameters).
  - *Task Arithmetic:* $\theta_{\mathrm{ta}}(\alpha) = \theta_0 + \alpha \cdot \frac{1}{N}\sum_{i=1}^{N}\tau_i$, with $\alpha \in \{0.5, 1.0\}$.
  - *TIES-Merging:* we keep top-$p$ per-task magnitudes, drop sign-conflicted coordinates, and then sum: $\theta_{\mathrm{ties}}(p) = \theta_{\mathrm{avg}} + \sum_{i=1}^{N}\mathcal{M}_p(\tau_i)$, with $p \in \{0.1, 0.3, 0.5, 0.7\}$.
  - *CART (origin-shifted low-rank):* we form $\theta_{\mathrm{avg}}$, shift origin, apply per-layer SVD truncation with rank $r$, and scale: $\theta_{\mathrm{cart}}(\eta, r) = \theta_{\mathrm{avg}} + \eta\,\hat{\tau}^{(r)}$, with $\eta \in \{0.5, 1.0\}$, $r \in \{0.1, 0.3, 0.5, 0.7\}$;
  - *$R^2$M-Merging(ours):* $\theta_{\mathrm{R^2M}}(\alpha, r) = \theta_0 + \tau_{\mathrm{core}} + \eta_{\mathrm{eff}}\,\tau_{merge}^{res}$,    $\alpha \in \{0.4, 0.5, 0.6\}$, $r \in \{0.1, 0.3, 0.5, 0.7\}$.

### B.2    TRAINING DETAILS

We strictly follow the official DF40 protocol without deviations in preprocessing, augmentation, optimization, or evaluation.

**Protocol.**    **Video I/O:** H.264 compression level `c23`; we sample **8 frames per clip** for both training and testing at **224×224** resolution. **Batching and system:** batch size 16 for training and testing; 1 GPU; `manualSeed=1024`. **Inputs:** no masks (`with_mask=false`) and no facial landmarks (`with_landmark=false`). **Normalization:** per-channel mean and standard deviation are set to (0.5, 0.5, 0.5).

**Data augmentation.**    Random horizontal flip ($p{=}0.5$); small rotations within $\pm10°$ ($p{=}0.5$); Gaussian blur with kernel size 3–7 ($p{=}0.5$); brightness and contrast jitter within $\pm0.1$ (each with $p{=}0.5$); JPEG quality jitter in [40, 100].

**Optimization.**    Adam by default with `lr`$=1 \times 10^{-5}$, $\beta_1{=}0.9$, $\beta_2{=}0.999$, $\epsilon{=}10^{-8}$, and weight decay $5\times10^{-4}$. When SGD is used, we set `lr`$=2 \times 10^{-4}$, momentum 0.9, and the same weight decay. We do not use a learning-rate scheduler. Training runs for **3 epochs**.

**Specialist task sets.**    **Task = 3** (three specialists; 24 total):

*FS (8):* `fsgan, faceswap, facedancer, blendface, simswap, mobileswap, e4s, inswap`.

*FR (8):* `MRAA, facevid2vid, fomm, sadtalker, hyperreenact, mcnet, one_shot_free, wav2lip`.

*EFS (8):* `SiT, ddpm, DiT, sd2.1, pixart, rddm, VQGAN, StyleGAN2`.

**Task = 6** (six specialists; 31 total):

*S1*: FS (5) = `uniface, simswap, mobileswap, faceswap, fsgan`.

*S2*: FS (4) = `inswap, blendface, e4s, facedancer`.

*S3*: FR (6) = `sadtalker, tpsm, fomm, MRAA, facevid2vid, pirender`.

*S4*: FR (6) = `hyperreenact, danet, lia, mcnet, one_shot_free, wav2lip`.

*S5*: EFS (5) = `pixart`, `StyleGANXL`, `StyleGAN3`, `DiT`, `ddpm`.

*S6*: EFS (5) = `rddm`, `StyleGAN2`, `SiT`, `VQGAN`, `sd2.1`.

**Details of the heterogeneous Real experiment (FF++ + CDF)   Datasets and Real/Fake composition.**

In addition to the three FF++-based specialists used in the main experiments (FS-ff, FR-ff, EFS-ff), we construct three further specialists whose Real data come from CDF (FS-cdf, FR-cdf, EFS-cdf). Since DF40 does not use CDF for training in its original protocol, we randomly split all available CDF videos into train/test/val sets for both Real and Fake, yielding:

- FSAll_Real: train/test/val = 1 282 / 320 / 1 602;    FSAll_Fake: 4 129 / 1 032 / 5 161.
- FRAll_Real: 1 709 / 427 / 2 136;    FRAll_Fake: 6 216 / 1 554 / 7 770.
- EFSAll_Real: 1 424 / 356 / 1 780;    EFSAll_Fake: 7 104 / 1 776 / 8 880.

Each CDF-based specialist (FS-cdf, FR-cdf, EFS-cdf) is trained on the corresponding split using exactly the same architecture, data augmentations, and optimization hyperparameters as the FF++-based specialists.

**Datasets and citations.**   `fsgan` (Nirkin et al., 2019), `faceswap` (fac, 2019), `simswap` (Chen et al., 2020), `facedancer` (Rosberg et al., 2023), `blendface` (Shiohara et al., 2023), `mobileswap` (Xu et al., 2022b), `e4s` (Liu et al., 2023c), `inswap` (ins). `uniface` (Xu et al., 2022a), `pirender` (Ren et al., 2021), `danet` (Hong et al., 2022), `lia` (Wang et al., 2022), `tpsm` (Zhao & Zhang, 2022), `MRAA` (Siarohin et al., 2021), `facevid2vid` (Wang et al., 2019), `fomm` (Siarohin et al., 2019), `sadtalker` (Zhang et al., 2023), `hyperreenact` (Bounareli et al., 2023), `mcnet` (Hong & Xu, 2023), `one_shot_free` (Wang et al., 2021), `wav2lip` (Prajwal et al., 2020). `VQGAN` (Esser et al., 2021), `StyleGAN2` (Karras et al., 2020), `StyleGAN3` (Karras et al., 2021), `StyleGANXL` (Sauer et al., 2022), `sd2.1` (Rombach et al., 2022b), `ddpm` (Ho et al., 2020), `rddm` (Liu et al., 2023a), `pixart` (Chen et al., 2024), `DiT` (Peebles & Xie, 2023), `SiT` (Atito et al., 2021).

## C   ADDITIONAL EXPERIMENTAL RESULTS

### C.1   EFFECT OF DIFFERENT LINEAR HEADS AFTER MERGING

In the main paper, we use a *single shared linear head* after merging, obtained by averaging the specialist heads. This design choice implicitly assumes that the merged embedding (Real core + Fake residuals) is the dominant factor, and that the exact choice of linear head has limited impact. To validate this assumption, we perform an ablation in which we keep the merged backbone fixed and vary only the linear head.

**Setup.**   We start from the $R^2M$-merged detector reported in Table 1 (Protocols 1 and 2). Let $h_{FS}$, $h_{FR}$, and $h_{EFS}$ denote the final linear heads of the FS, FR, and EFS specialists, respectively. We attach each of these heads to the *same* $R^2M$-merged backbone (without any further training) and compare their AUCs with the shared (averaged) head used in the main paper. Across both Protocol 1 (FF++ Real) and Protocol 2 (CDF Real), the average AUC differences between the shared head and the three specialist heads are below $0.002$, confirming that $R^2M$'s gains mainly come from the merged embedding rather than the specific choice of linear head.

**Results on seen domains (FF++ and CDF).**   Table A.1 shows the AUCs on FF++ and CDF domains. The three heads yield almost identical performance on top of the merged backbone: across six domains, differences between $h_{FS}$, $h_{FR}$, and $h_{EFS}$ are on the order of $10^{-3}$, and their FF++/CDF means differ by at most $1 \times 10^{-3}$. This supports our claim that the merged embedding learned by $R^2M$ dominates performance, and that the precise choice of head has only a minor effect.

**Discussion.**   Across both protocol 1 and protocol 2, swapping the FS/FR/EFS heads on top of the same merged backbone yields only marginal differences in AUC (typically $\leq 0.001$). This empirically justifies our design choice in the main paper: $R^2M$'s benefit comes primarily from the *merged*

Table A.1: **Linear head ablation on seen domains.** AUC on three FF++ domains (FS-FF, FR-FF, EFS-FF) and three CDF domains (FS-CDF, FR-CDF, EFS-CDF). "Avg. head" denotes the used average head in the main paper. The remaining rows correspond to attaching the FS/FR/EFS specialist heads to the same $R^2M$-merged backbone without further training.

| | FF++ domains | | | | CDF domains | | | |
|---|---|---|---|---|---|---|---|---|
| | FS-FF | FR-FF | EFS-FF | Mean | FS-CDF | FR-CDF | EFS-CDF | Mean |
| Avg. head | 0.977 | 0.992 | 0.996 | 0.988 | 0.902 | 0.912 | 0.942 | 0.919 |
| FS head | 0.9771 | 0.9921 | 0.9953 | 0.9882 | 0.9021 | 0.9117 | 0.9423 | 0.9187 |
| FR head | 0.9763 | 0.9923 | 0.9963 | 0.9883 | 0.9017 | 0.9119 | 0.9418 | 0.9185 |
| EFS head | 0.9769 | 0.9921 | 0.9954 | 0.9881 | 0.9014 | 0.9120 | 0.9416 | 0.9183 |

*embedding* (shared Real core + Fake residuals), and a simple shared linear head is sufficient in practice.

## C.2 NUMERICAL VALUES FOR DROP–GAIN ANALYSIS (FIGS.4)

Figs. 4 in the main text visualize the trade-off between (i) performance retention on seen tasks and (ii) performance gain on unseen forgeries, using heatmaps. For completeness and to facilitate precise comparison across methods, we report in this section the exact numerical values underlying those figures.

Table A.2 lists these drops for all merging methods shown in Fig. 4a.

Table A.2: **Numerical values for Fig. 4a (Seen Drop heatmap).** Drop is defined as *best specialist AUC − merged AUC*; lower is better. Rows correspond to (task, domain) pairs and columns to merging methods.

| Task–domain | All-in-one | Weight Avg. | Task Arith. | TIES-Merging | CART | $R^2M$ (ours) |
|---|---|---|---|---|---|---|
| FS-ff | 0.033 | 0.027 | 0.039 | 0.036 | 0.019 | 0.018 |
| FR-ff | 0.002 | 0.002 | 0.004 | 0.006 | 0.005 | 0.007 |
| EFS-ff | 0.021 | 0.017 | 0.034 | 0.038 | 0.004 | 0.003 |
| FS-cdf | 0.200 | 0.134 | 0.218 | 0.140 | 0.108 | 0.057 |
| FR-cdf | 0.055 | 0.006 | 0.027 | 0.017 | 0.041 | 0.003 |
| EFS-cdf | 0.623 | 0.222 | 0.063 | 0.098 | 0.082 | 0.047 |

Table A.3 reports these gains for all methods in Fig. 4b.

Table A.3: **Numerical values for Fig. 4b (Unseen Gain heatmap).** Gain is defined as *merged AUC − best specialist AUC*; higher is better. Rows correspond to unseen forgery generators and columns to merging methods.

| Forgery | All-in-one | Weight Avg. | Task Arith. | TIES-Merging | CART | $R^2M$ (ours) |
|---|---|---|---|---|---|---|
| DeepFaceLab | -0.008 | 0.038 | -0.005 | 0.005 | 0.051 | 0.054 |
| HeyGen | -0.012 | 0.053 | -0.032 | 0.006 | 0.035 | 0.044 |
| MidJourney | -0.426 | 0.010 | 0.092 | -0.058 | -0.044 | -0.052 |
| WhichIsReal | -0.182 | -0.239 | -0.439 | -0.367 | -0.197 | -0.202 |
| StarGAN | -0.041 | 0.052 | -0.010 | 0.011 | 0.074 | 0.072 |
| StarGANv2 | -0.063 | -0.049 | -0.198 | -0.179 | 0.003 | 0.001 |
| StyleCLIP | -0.191 | -0.317 | -0.448 | -0.284 | -0.139 | -0.092 |
| CollabDiff | -0.143 | -0.149 | -0.375 | -0.370 | -0.042 | -0.023 |

These tables make explicit that $R^2M$ attains the smallest mean seen drop and the highest (or near-highest) gains on many unseen generators, while degrading more gracefully than other merging baselines when performance drops are unavoidable.

## C.3  Task-3 vs. Task-6 configurations (Figs.6)

In the main text, we compare two $R^2M$ configurations (Task-3 vs. Task-6. Both configurations are evaluated on the six seen settings from Table 1 (FS/FR/EFS under Protocols 1 and 2) and on the eight unseen generators from Protocol 3. The difference between Task-3 and Task-6 lies only in how we structure the merge in parameter space: Task-3 uses three residual directions (one per forgery family: FS, FR, EFS), whereas Task-6 refines them into six residual directions by splitting each family into two sub-groups (e.g., two FS-style subgroups, two FR-style, two EFS-style), while keeping the evaluation data fixed. Supplementary grouped dumbbell plots (Figs. 6a and 6b) visualize these comparisons by connecting, for each method, its AUC under Task-3 (circle) and Task-6 (square).

**Seen settings (Protocols 1 & 2).**   Table A.4 reports, for each of the six seen evaluation settings from Table 1, the AUCs of four merging baselines under Task-3 and Task-6. These are exactly the values used to draw the grouped dumbbells on seen data.

Table A.4: **Task-3 vs. Task-6 on seen evaluation settings.** AUC on the six seen settings (FS/FR/EFS under Protocols 1 and 2) for four merging methods under Task-3 and Task-6. The evaluation data are identical across Task-3 and Task-6; only the merge configuration (3 residuals vs. 6 residuals) differs. These values correspond to the segments in the seen grouped dumbbell plot.

|                    | Weight Averaging | | Task Arithmetic | | CART | | $R^2M$ (ours) | |
|--------------------|-------|-------|-------|-------|-------|-------|-------|-------|
| Seen setting       | T3    | T6    | T3    | T6    | T3    | T6    | T3    | T6    |
| (1) FS, Protocol 1 | 0.968 | 0.956 | 0.956 | 0.506 | 0.976 | 0.978 | 0.977 | 0.981 |
| (2) FR, Protocol 1 | 0.997 | 0.996 | 0.995 | 0.570 | 0.994 | 0.997 | 0.992 | 0.998 |
| (3) EFS, Protocol 1| 0.982 | 0.971 | 0.965 | 0.558 | 0.995 | 0.964 | 0.996 | 0.978 |
| (4) FS, Protocol 2 | 0.825 | 0.641 | 0.741 | 0.569 | 0.851 | 0.707 | 0.902 | 0.789 |
| (5) FR, Protocol 2 | 0.909 | 0.883 | 0.888 | 0.637 | 0.874 | 0.855 | 0.912 | 0.902 |
| (6) EFS, Protocol 2| 0.767 | 0.413 | 0.926 | 0.569 | 0.907 | 0.325 | 0.942 | 0.723 |

**Unseen forgeries (Protocol 3).**   For the eight unseen generators (DeepFaceLab, HeyGen, Mid-Journey, WhichIsReal, StarGAN, StarGANv2, StyleCLIP, CollabDiff), we summarize the mean AUC across all generators for each method in Table A.5. These are the values used for the bold "mean-only" dumbbells in the unseen plot.

Table A.5: **Task-3 vs. Task-6 on unseen forgeries (mean AUC).** Mean AUC across 8 unseen generators for four merging methods under Task-3 and Task-6. The evaluation protocol (unseen generators) is fixed; only the merge configuration differs.

| Method           | Task-3 mean | Task-6 mean |
|------------------|-------------|-------------|
| Weight Averaging | 0.724       | 0.684       |
| Task Arithmetic  | 0.622       | 0.553       |
| CART             | 0.766       | 0.660       |
| $R^2M$ (ours)    | 0.774       | 0.771       |

## C.4  Ablations of $R^2M$ Components

We ablate the two scalar knobs in equation 6 the residual rank ratio $r$ (fractional SVD rank for the residual) and the merge-strength scale $\alpha$ in the norm-normalized $\eta$. Across $\alpha \in \{0.4, 0.5, 0.6\}$, increasing $r$ consistently improves the Protocol 1 (FF) AUCs for all categories (FS/FR/EFS) until a clear plateau around $r \in [0.5, 0.7]$ (Fig.A.1). In particular, the configuration $\alpha{=}0.5$ with $r \in [0.5, 0.7]$ yields FF mean AUCs of 0.987–0.987, matching the best settings while avoiding the instability observed at lower ranks. We therefore fix this setting when evaluating on Protocol 2 (CDF) and on Protocol 3 (unseen).

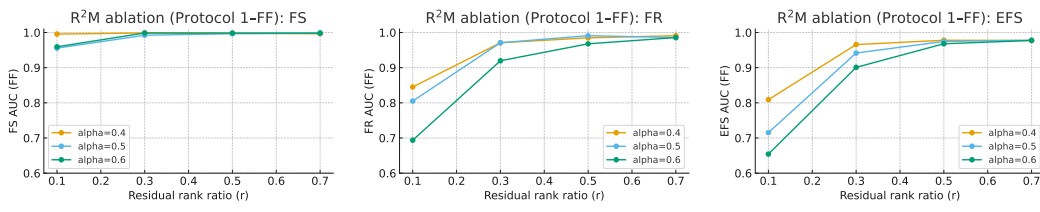

(a) FS (Protocol 1–FF)      (b) FR (Protocol 1–FF)      (c) EFS (Protocol 1–FF)

Figure A.1: **$R^2M$ ablation on residual rank $r$ and merge-strength scale $\alpha$.** Each curve varies $r$ for a fixed $\alpha \in \{0.4, 0.5, 0.6\}$; y-axis shows per-category AUC on FF.

**Trend w.r.t. rank ($r$).** Prior work (*e.g.*, CART) notes that increasing the SVD rank can amplify cross-task interference and hurt generalization. In our setting, however, we first estimate a shared *Real core* from the top-$k$ directions of the *centered* task matrix and *add* this core, while the residual path is built around the averaged origin and aggregated with norm-normalized scaling. This centering-and-scaling design cancels much of the destructive across-task drift before any truncation, so raising the residual rank exposes additional informative variation rather than amplifying interference. Empirically, increasing $r$ to $0.5\sim0.7$ improves AUC without instability, indicating that useful residual structure is recovered while nuisance coupling remains controlled.

**Trend w.r.t. merge strength ($\alpha$).** The norm-normalized scaling of $\eta$ reduces sensitivity to the absolute magnitudes of the core and residual updates. When the residual rank is *small* ($r \in \{0.1, 0.3\}$), **smaller** $\alpha$ is preferable: stronger scaling can over-amplify a too-low-rank residual and slightly hurt performance (e.g., at $r{=}0.1$ the FF mean AUC is higher with $\alpha{=}0.4$ than with $\alpha{=}0.6$). As the residual rank increases, this dependence diminishes: once $r \geq 0.5$, the curves largely *saturate* and the gap between $\alpha{=}0.4, 0.5, 0.6$ becomes negligible. We therefore adopt a robust operating point around $\alpha{=}0.5$ with $r{=}0.7$ for CDF and unseen evaluations.

