# OpenReview forum: "Real-Aware Residual Model Merging for Deepfake Detection"
_ICLR.cc/2026/Conference — Submitted to ICLR 2026_

### Official Review · Reviewer_9CNL · 2025-10-23

**Soundness:** 3
**Presentation:** 3
**Contribution:** 3
**Rating:** 6
**Confidence:** 4

**Summary:**

This paper proposes the first method that introduces model merging to the deepfake detection task. It analyzed the real similarity and fake distinction among the DF40 dataset, and then proposed R2M with a theoretical explanation. The results show its effectiveness in balancing different types of forgery methods in comparison with the specialist.

**Strengths:**

1.	The paper introduces model merging into deepfake detection for the first time.
2.	It provides a reasonable analysis of real–fake similarity on the DF40 dataset.
3.	The proposed R2M method shows balanced performance with theoretical support.

**Weaknesses:**

1.	Since DF40 is used, all real samples come from the same FF++ real, so it is unsurprising that they share the same distribution. However, this also limits the applicability of the conclusion. What if the training data contain reals from different distributions, e.g., FF++ and CDF?
2.	Regarding generalization: while it is understandable that model merging can improve in-domain performance, it remains unclear why merging would enhance cross-domain generalization. This point requires more explanation.
3.	In Table 1: The experimental results appear unusual — the All-in-one model fails completely on EFS detection, and even reverses predictions. Why would changing the real distribution cause previously learned forgery types to invert their detection behavior? This phenomenon warrants deeper analysis, particularly to clarify how model merging could lead to label inversion.
4.	More experiments on generalization are needed, for example, by evaluating on additional datasets (e.g., DFDCP, DFD) and comparing with more generalization-oriented methods (e.g., [1] and [2]).
5.	The writing requires further proofreading. For instance, DF40 is incorrectly cited — it is not (Qian et al., 2024) but rather (Yan et al., 2024).
6.	Within the same forgery category, the fake cases also require further analysis to validate the similarity findings in Fig. 2 — for example, within EFS. It would be helpful to train specialist models separately using DiT, SiT, and StyleGAN to support this analysis.

[1] Effort: Efficient orthogonal modeling for generalizable ai-generated image detection //ICML’25

[2] Can we leave deepfake data behind in training deepfake detector? //NIPS'24

**Questions:**

Please refer to the weaknesses.

---

> ### Author Response · Authors · 2025-11-21
> **1st response**
>
> We thank the reviewer for the positive overall assessment and for raising several important points about Real diversity, generalization, and within-category analysis.
>
> ### **Real from multiple distributions (FF + CDF)**
>
> We agree that, in the original DF40 setting where all reals come from FF, it is unsurprising that Real features appear similar. To test the robustness of our conclusion, we have trained **three additional specialists on CDF real** (FS-cdf, FR-cdf, EFS-cdf) in addition to the original three FF specialists (FS-ff, FR-ff, EFS-ff), and merged all six in the Sec 4.4 of revised manuscript.
>
> | Model | FS-ff | FR-ff | EFS-ff | FS-cdf | FR-cdf | EFS-cdf |
> | --- | --- | --- | --- | --- | --- | --- |
> | FS-ff | 0.995 | 0.912 | 0.766 | 0.959 | 0.696 | 0.621 |
> | FR-ff | 0.923 | 0.9996 | 0.742 | 0.503 | 0.923 | 0.106 |
> | EFS-ff | 0.676 | 0.814 | 0.999 | 0.635 | 0.608 | 0.988 |
> | FS-cdf | 0.754 | 0.647 | 0.616 | 0.9999 | 0.989 | 0.982 |
> | FR-cdf | 0.557 | 0.849 | 0.532 | 0.991 | 0.9999 | 0.929 |
> | EFS-cdf | 0.329 | 0.244 | 0.923 | 0.955 | 0.961 | 0.9999 |
> | CART | 0.867 | 0.957 | 0.954 | 0.998 | 0.994 | 0.996 |
> | R²M | 0.901 | 0.969 | 0.962 | 0.996 | 0.994 | 0.996 |
>
> As expected, asking a single model to cover six specialists trained on two different real distributions is substantially more challenging than the original FF++-only case.
> Nevertheless, $R^{2}$M consistently achieves higher AUC than CART on the harder FF++ domains, while matching CART on the saturated CDF domains.
> This indicates that the SVD-based Real core in $R^{2}$M continues to provide a stabilizing shared subspace even when Real comes from multiple datasets, and that $R^{2}$M \emph{degrades more gracefully} than CART as real diversity increases.
>
> In this more challenging setting, a single model must cover Real from both FF and CDF. CART  lose some performance relative to individual specialists, but R2M still **outperforms CART on the FF domains** while keeping CDF performance near saturation. This shows that even when Real comes from two distinct datasets, the SVD over task vectors still finds a meaningful shared Real subspace and yields more robust merging than CART.
>
> ### **Why merging helps generalization**
>
> In the revised paper, we clarify that R2M improves cross-domain performance mainly by:
>
> - **combining complementary Fake residuals** from different specialists (each capturing distinct artifacts), and
> - **stabilizing the Real side** via a shared Real core, which prevents the decision boundary from being dominated by a single dataset or forgery family.
>
> Empirically, this appears as **better average and worst-case AUC** across unseen generators (DeepFaceLab, HeyGen, MidJourney, StyleClip, CollabDiff, etc.) compared to single specialists and simple averaging. Intuitively, the Real core **“lifts the floor”** of performance: even when the task becomes harder due to diversity, R2M keeps the minimum performance across domains higher than alternative merges.
>
> At the same time, we emphasize that our primary goal is not to compete with specialized “zero-shot generalization” methods, but to provide a **fast adaptation mechanism**: once a new forgery family is observed, it can be turned into a specialist and integrated via R2M to restore stable performance quickly.
>
> ### **All-in-one failure and apparent label inversion on EFS**
>
> We agree that the All-in-one baseline’s behavior on EFS (AUC < 0.5) looks surprising. This inversion arises because the All-in-one model is trained as a single binary classifier on the union of all data. When we change the real distribution (e.g., to CDF-only real), the model overfits to dataset-specific statistics. For EFS, whose distribution differs substantially between FF and CDF, most EFS-CDF fakes can end up on the “Real” side of this biased boundary, leading to inverted predictions—even though EFS-FF was previously handled reasonably well by some specialists.
>
> In contrast, R2M explicitly decomposes the specialists into a shared **Real core** and **generator-specific Fake residuals**. This separation prevents the merged detector from collapsing onto a single real domain: the Real core stabilizes the real representation, while the Fake residuals preserve generator-specific cues. Empirically, when we switch the real distribution (protocol 2), R2M does not exhibit such inversion on EFS, whereas only the All-in-one model does. We have clarified this explanation in the revised manuscript and highlight it as an example of why a structured merging strategy is preferable to a monolithic All-in-one baseline under real-distribution shifts.

---

> ### Author Response · Authors · 2025-11-21
> **2nd response**
>
> ### **Additional generalization experiments and comparison with [1,2]**
>
> As noted in the response to Reviewer 2, methods such as Effort [1] and “Can we leave deepfake data behind” [2] are **training-time generalization methods for a single detector**, whereas R2M is a **training-free merging mechanism** that can be applied on top of any such detector. In the revised related-work section, we discuss this complementary relationship and point out that R2M could in principle merge specialists that were themselves trained with [1] or [2].
>
> ### **Writing corrections**
>
> We thank the reviewer for pointing out that DF40 should be cited as **(Yan et al., 2024)** rather than (Qian et al., 2024). We have corrected this and carefully proofread the manuscript for other typos and inconsistent references.
>
> ### **Within-category fake analysis**
>
> We appreciate the reviewer’s suggestion to further analyze similarity within a forgery category, for example by training separate specialists on DiT, SiT, and StyleGAN inside EFS. Training fully independent specialists for each individual generator, however, would require a substantial amount of additional computation. Instead, we took advantage of our existing \emph{Task = 6} configuration, where each family (FS, FR, EFS) is already split into two sub-groups of generators:
> S1/S2 for FS, S3/S4 for FR, and S5/S6 for EFS (see Appendix B.2 for the exact lists, which include DiT, SiT, and StyleGAN variants). We then measured how the cosine similarity between the WA(Weight Average) and each group specialist changes for different fake families.
> The summary in the table below shows that, for each family, the largest decrease in similarity always occurs on its own fakes:
> – For FS, the similarity on Fake_fs decreases by −4.82%, while the changes on Fake_fr and Fake_efs are much smaller (+0.13% and +2.18%).
> – For FR, all changes are relatively small, but the decrease is still concentrated on Fake_fr (-0.41% ), indicating that the FR-specific representation is the most affected.
> – For EFS, the effect is strongest: the similarity on Fake_efs drops sharply by −18.04%, whereas Fake_fs and Fake_fr change by only −1.94% and −3.49%.
> Together with the fact that Real similarity remains high across all splits (0.75,0.72, 0.75), this confirms our claim: the Real component is shared and stable, while the representation of a family’s own fakes is much more sensitive to which generators are included, even within the same category.
>
> | Family | Δ Fake_fs (%) | Δ Fake_fr (%) | Δ Fake_efs (%) |
> |--------|---------------|---------------|----------------|
> | FS     |     -4.82     |     +0.13     |     +2.18      |
> | FR     |     +1.33     |     -0.41     |     +1.67      |
> | EFS    |     -1.94     |     -3.49     |    -18.04      |
>
> This within-category analysis shows that, even when we refine a broad category (e.g., EFS) into finer generator sub-groups, the shared Real structure persists while the generator-specific Fake representation is much more sensitive to which generators are included.
> We agree that a fully \emph{per-generator} analysis (e.g., diffusions/GANs specialists) would provide an even more granular view, and we regard this as a promising extension beyond the current computational budget.

---

### Official Review · Reviewer_3mcr · 2025-10-26

**Soundness:** 3
**Presentation:** 3
**Contribution:** 3
**Rating:** 6
**Confidence:** 4

**Summary:**

This paper proposes a training-free model merging approach that enhances the generalization ability of deepfake detectors by combining specialist parameters trained on different forgery types. Specifically, the authors argue that the “real” class direction can be well preserved during merging, while class-specific forgery artifacts are suppressed. To this end, they employ SVD decomposition to extract dominant directions and obtain a generalizable merged model.

Extensive experiments on the DF40 dataset demonstrate that the proposed method largely retains each expert’s detection capability. The authors further claim that their method exhibits high scalability.

**Strengths:**

1. This paper introduces a valuable setting: Merging detectors via a training-free approach to obtain generalized deepfake detectors at low cost.

2.  The authors provide rich theoretical justifications supporting the effectiveness of their proposed method.

3. The proposed approach is extensively evaluated under multiple protocols, showing competitive or superior AUC retention on seen tasks and improved generalization to unseen forgeries.

**Weaknesses:**

1. Although the authors have discussed using a single averaged linear head after merging is enough to get the results, empirical validation is missing. The authors are encouraged to compare the following heads: i) Averaged linear head; ii) Specialist-specific linear heads; iii) Re-tuned linear head (after merging).

2. Theoretical analysis relies on several assumptions: i) Local linearity; ii) Bounded remainder terms; and iii) Mild spectral gap. These assumptions may fail when specialist models differ substantially or when the networks exhibit high nonlinearity. Consequently, the validity of the core theoretical result (Proposition 1) hinges on whether such local properties still hold in large-scale models.

3. The proposed approach shows scalability with six models. However, as the diversity of deepfake generation methods continues to grow, the current scale remains modest. It is unclear how the method performs when the number of experts increases dramatically (dozens or hundreds), especially when specialists have uneven capabilities or have been trained on partially overlapping domains. Since the proposed method depends on low-rank SVD, the dominance of a shared “real” direction may diminish as task vectors diversify.

4. This paper argues for the asymmetry between shared “Real” and generator-specific “Fake” features. Nevertheless, it does not explore scenarios where the real data distribution changes or diversifies. For instance, when real samples come from new domains (e.g., different camera sensors or capture conditions). In such cases, the assumption of a single, stable “Real” direction may no longer hold.

5. Merging parameters across different specialists could introduce vulnerabilities such as trojan signatures or model poisoning. The authors  are suggested to perform a security analysis to R2M robustness in this regard.

**Questions:**

Please see Weaknesses

---

> ### Author Response · Authors · 2025-11-21
> **1st response**
>
> We thank the reviewer for the insightful comments on theory, scalability, and security.
>
> ### **Linear heads after merging: averaged vs specialist heads**
>
> We appreciate the suggestion to empirically validate our claim that a single shared linear head suffices after merging. In the revised manuscript, we report an additional experiment in Sec.C.1, with the merged backbone fixed, we compare:
>
> 1. the **averaged/shared head** used in the paper, and
> 2. the **specialist-specific heads** (FS, FR, EFS) applied on the merged backbone.
>
> Across 6 seen protocols (FS/FR/EFS on FF and CDF) , all variants produce **almost identical** AUCs; differences are on the order of 0.001. This supports our claim that **the merged embedding (Real core + Fake residuals) is the dominant factor**, and the exact linear head plays only a minor role. We summarize these numbers in an appendix section C.1 and table A.1.
>
> | Head | FS-FF | FR-FF | EFS-FF | Mean (FF++) | FS-CDF | FR-CDF | EFS-CDF | Mean (CDF) |
> | --- | --- | --- | --- | --- | --- | --- | --- | --- |
> | Avg. head | 0.977 | 0.992 | 0.996 | 0.988 | 0.902 | 0.912 | 0.942 | 0.919 |
> | FS head | 0.9771 | 0.9921 | 0.9953 | 0.9882 | 0.9021 | 0.9117 | 0.9423 | 0.9187 |
> | FR head | 0.9763 | 0.9923 | 0.9963 | 0.9883 | 0.9017 | 0.9119 | 0.9418 | 0.9185 |
> | EFS head | 0.9769 | 0.9921 | 0.9954 | 0.9881 | 0.9014 | 0.9120 | 0.9416 | 0.9183 |
>
> ### **On the theoretical assumptions**
>
> We agree that Proposition 1 relies on local assumptions: local linearity, bounded higher-order terms, and a mild spectral gap. Our aim is to give a **local mechanistic explanation** in the regime where:
>
> - all specialists are fine-tuned from the **same pre-trained backbone**,
> - with similar architectures and moderate parameter drift.
>
> This is precisely the practical setting of our experiments. In the revised paper, we clarify that:
>
> - the theorem should be interpreted as a **local approximation** around a common initialization, and
> - when specialists differ drastically (e.g., trained from scratch or with very different architectures), these assumptions may fail and the theory should be read as heuristic motivation rather than a global guarantee.
>
> We also highlight extending the analysis to more diverse and highly nonlinear regimes as an important direction for future work.
>
> ### **Scalability and many/overlapping specialists**
>
> We agree that scalability and interference across tasks are central issues in model merging, especially as the number of specialists grows. In deepfake detection, datasets like DF40 already group past forgeries into broad categories (FS, FR, EFS). Our design philosophy is to keep the number of **high-level categories** moderate (by grouping older methods) and use R2M to absorb **new** methods as they appear.
>
> The new Uniface experiment (Sec.4.3) illustrates this: when a new FS-style generator appears, the existing merged model already performs well; if necessary, we can train a Uniface specialist and merge it with minimal effect on previous domains. For well-separated new families (e.g., MidJourney content), we can introduce a new specialist and treat it as a new category.
>
> We believe this naturally leads to a **hierarchical merging** view in future work:
>
> (i) new specialists are monitored; if they behave similarly to an existing category, they are merged into that category’s core;
>
> (ii) if they represent a qualitatively new family, they initialize a new core;
>
> (iii) higher-level merging across cores can then be applied. While this goes beyond the scope of the current paper, our results show that R2M already works well for the “first layer” of such a hierarchy, and we explicitly mention this recursive, self-expanding extension as future work.

---

> ### Author Response · Authors · 2025-11-21
> **2nd response**
>
> ### **When Real distribution changes or diversifies**
>
> The reviewer correctly notes that our Real/Fake asymmetry may weaken when Real becomes strongly multi-domain. Our new 6-specialist FF + CDF experiment in Section 4.4 is a first step toward this scenario. Real now comes from two datasets with different capture conditions, making the merging problem inherently harder. In this regime, **all merged models degrade** relative to individual specialists, which is unavoidable, but R2M still **degrades more gracefully than CART**: it achieves higher AUC on the more challenging FF domains while matching CART on the saturated CDF domains.
>
> | Model | FS-ff | FR-ff | EFS-ff | FS-cdf | FR-cdf | EFS-cdf |
> | --- | --- | --- | --- | --- | --- | --- |
> | FS-ff | 0.995 | 0.912 | 0.766 | 0.959 | 0.696 | 0.621 |
> | FR-ff | 0.923 | 0.9996 | 0.742 | 0.503 | 0.923 | 0.106 |
> | EFS-ff | 0.676 | 0.814 | 0.999 | 0.635 | 0.608 | 0.988 |
> | FS-cdf | 0.754 | 0.647 | 0.616 | 0.9999 | 0.989 | 0.982 |
> | FR-cdf | 0.557 | 0.849 | 0.532 | 0.991 | 0.9999 | 0.929 |
> | EFS-cdf | 0.329 | 0.244 | 0.923 | 0.955 | 0.961 | 0.9999 |
> | CART | 0.867 | 0.957 | 0.954 | 0.998 | 0.994 | 0.996 |
> | R²M | 0.901 | 0.969 | 0.962 | 0.996 | 0.994 | 0.996 |
>
> As expected, asking a single model to cover six specialists trained on two different real distributions is substantially more challenging than the original FF++-only case.
> Nevertheless, $R^{2}$M consistently achieves higher AUC than CART on the harder FF++ domains, while matching CART on the saturated CDF domains.
> This indicates that the SVD-based Real core in $R^{2}$M continues to provide a stabilizing shared subspace even when Real comes from multiple datasets, and that $R^{2}$M \emph{degrades more gracefully} than CART as real diversity increases.
>
> We emphasize this in the revised text: the goal is not to claim that performance never drops as Real diversity increases, but that $R^{2}$M **maintains a stronger “floor” of performance than alternative merging strategies** when diversity grows.
>
> ### **Security aspects: trojans and poisoning**
>
> We appreciate the suggestion to consider trojans and model poisoning. Our current work assumes a **trusted training pipeline**: all specialists are trained by the same party on clean data. We did not explicitly study adversarially poisoned or backdoored specialists, and we now state this assumption clearly in the revised manuscript.
>
> We also add a short discussion of potential risks when merging untrusted models and point to possible mitigations—such as validating each specialist on held-out clean data, examining residuals for anomalous behavior, or integrating  $R^{2}$M with existing trojan-detection techniques—as important directions for future work.

---

### Official Review · Reviewer_8SSQ · 2025-10-31

**Soundness:** 3
**Presentation:** 3
**Contribution:** 3
**Rating:** 4
**Confidence:** 5

**Summary:**

This paper introduces **R²M**, a novel method for merging expert models in the domain of Deepfake detection. The core idea is to leverage SVD to decompose task vectors, treating shared "Real" features as a principal component and generator-specific "Fake" artifacts as residuals. This allows for a training-free and efficient merging process without a drop in performance. However, I have some concerns regarding the method's practical utility in real-world scenarios and its overall impact on advancing the field.

**Strengths:**

1. **Well Motivation:** The paper's motivation is clear and reasonable. Figure 2 provides a compelling and intuitive illustration of the core hypothesis: different specialist models share a common understanding of "Real" data while diverging in their representation of "Fake" data.

2. **Efficient and Effective Method:** The proposed R²M method is elegant in its simplicity. Being training-free, it offers a highly efficient way to combine specialists, and the experiments show that it does so with negligible performance degradation on seen tasks.

**Weaknesses:**

1. **Comparison of Similar SVD Technique Used for Detection is Missing:** Previous work like Effort (ICML'25) also proposed the similar  SVD-based approach for improved detection performance. More discussion and comparison for similar methods are needed.

2. **Clarity of Visualizations:** The readability of several experimental figures is a concern. In Figure 3 (heatmaps) and Figure 5 (dumbbell plots), it is difficult to discern the precise numerical gains or losses.

3. **Marginal Performance Gains:** The improvement of R²M over prior model merging techniques, like CART, appears to be marginal.

4. **Concerns about Practical Utility and Scalability:** My primary concern lies with the practical application of the proposed merging strategy. The experiments partition domains based on broad forgery categories (e.g., FS, FR, EFS) rather than specific generator models (e.g., FSGAN, FaceSwapV2). This raises a crucial question: how should the framework handle the incremental addition of a new forgery method that belongs to an existing broad category? If a new FaceSwap variant emerges, would one retrain the entire FS specialist, or would a new, more granular merging strategy be required? The current setup does not seem to address this realistic scenario.

5. **Strange Performance of Comparison Methods:** For instance, the performance of *Specialist-FR* on EFS samples is so low (AUC=0.099). Can the author explain?

**Questions:**

1. **Fine-tuning Details:** For reproducibility and clarity, it would be beneficial to specify which parameters of the backbone were fine-tuned for the specialist models. For instance, was it only the final linear layer, or were other parts of the network also updated?

2. **Lack of Pipeline Diagram:** It would be better to provide a high-level pipeline diagram. A clear visual representation of the R²M process—from task vectors to the final merged model—would significantly aid reader comprehension.

3. **Ambiguity of "All-in-one" Baseline:** The training setup for the "All-in-one" baseline is unclear. Was it trained as a single binary (Real vs. Fake) classifier on all forgery data? To further strengthen the experimental results, I would suggest including a baseline where the "All-in-one" model is trained on a multi-class Fake detection task (i.e., classifying each specific forgery type), with the logits for all fake classes then aggregated for binary evaluation. This would provide a more robust comparison.

4. **Limited Impact on the Deepfake Detection Field:** While the authors position the work as a contribution to model merging, its tangible impact on advancing the core challenges in Deepfake detection itself seems limited. The problem of generalizing to unseen forgery families, a critical issue in the field, is not substantially improved by this merging approach.

---

> ### Author Response · Authors · 2025-11-21
> **1st response**
>
> We thank the reviewer for the detailed feedback and for highlighting both the strengths and concerns about practicality and impact.
>
> ### **Comparison of Similar SVD Technique Used for Detection is Missing**
>
> We appreciate the pointer to Effort  (ICML’25). Effort also employs SVD to decompose parameters into “semantic” and “forgery” components, but its **goal and usage differ from ours**:
>
> - Effort is a **training-time method** that improves **generalization of a single detector**, by guiding how that detector is trained.
> - R2M is a **training-free model-merging framework** that starts from **multiple already-trained specialists** and decomposes their **task vectors across models** into a shared Real direction and generator-specific Fake residuals, which are then recombined.
>
> Thus, Effort is a detection/generalization technique, whereas R2M is about **how to merge specialists in parameter space**. However, the two approaches are **complementary**: specialists could first be trained using Effort (or any other advanced objective), and then merged via R2M. We have made this relationship explicit in the revised related-work section, highlighting both conceptual similarities (SVD-style decomposition, preservation of semantic/Real information) and differences (training vs. merging).
>
> ### **Clarity of Visualizations**
>
> We agree that Figs. 3 (heatmaps) and 5 (dumbbell plots) are difficult to read numerically. In the revised version, we have: added a **supplementary table** listing all underlying AUC values, so that precise gains/losses can be read unambiguously. We explicitly refer to this table in the figure captions in C.2 and C.3.
>
> ### **Magnitude of gains over CART and when they matter**
>
> We acknowledge that in some main-table settings, the average improvement over CART appears modest. Our design goals are **training-free operation, stability, and preservation of specialist performance**. In that regime, even small but **consistent** improvements over strong baselines such as CART are meaningful, especially with zero additional training cost.
>
> Moreover, as Real distributions become more heterogeneous, the benefit of modeling a shared Real core becomes more visible. In our new 6-specialist experiment in Sec.4.4 mixing FF and CDF (described in the section ), R2M improves FF performance noticeably over CART while keeping CDF saturated. This indicates that R2M **degrades more gracefully** than CART as Real becomes diverse, which is precisely the challenging regime for deployment.
>
> ### **Practical utility and incremental addition of new generators**
>
> We fully agree that incremental addition of new forgery methods inside an existing category is realistic. To address this, we have conducted a new experiment and added it in Sec 4.3 case 1:
>
> | Merged specialists | FS | FR | EFS | Uniface |
> | --- | --- | --- | --- | --- |
> | FS + FR + EFS | 0.977 | 0.992 | 0.996 | 0.972 |
> | FS + FR + EFS + Uniface | 0.976 | 0.991 | 0.996 | 0.989 |
>
> - the original FS specialist is trained on eight FS-style generators;
> - the merged model (FS+FR+EFS) already achieves **0.972 AUC** on a new FS-style dataset **Uniface**, even though no Uniface specialist exists;
> - we then train a dedicated **Uniface specialist** and merge **four** specialists (FS, FR, EFS, Uniface) with R2M.
>
> After merging four specialists, **Uniface AUC increases from 0.972 to 0.989**, while performance on FS/FR/EFS remains almost unchanged (differences ≤ 0.001–0.002). This shows that:
>
> - if a new generator is similar to an existing family, R2M can already handle it reasonably well **without** a new specialist;
> - once a new specialist is available, R2M can integrate it **incrementally** with negligible interference to existing domains.
>
> We have added this experiment and analysis to the Section 4.3. We also note in the discussion that if one keeps adding many specialists, it becomes natural to adopt a **hierarchical merging** strategy (e.g., merging closely related specialists into sub-cores, then merging these cores), which we view as a promising direction for future work.
>
> ### **“Strange” FR performance on EFS**
>
> The very low FR-on-EFS AUC reflects a **strong domain gap**, rather than an implementation error. Even in the DF40 paper, FR-only CLIP models perform poorly on entire-face synthesis from other datasets (AUC around 0.2). Our FR specialist is trained only on 8 FR manipulations, using a slightly reduced subset to balance the number of generators across families; as a result, its cross-family performance on EFS is even lower, but still consistent with a severely mismatched domain. We do not consider this behavior “strange”; rather, it illustrates how narrowly trained specialists can fail on very different forgery families and motivates the need for robust merging.

---

> ### Author Response · Authors · 2025-11-21
> **2nd response**
>
> ### **Answers to specific questions**
>
> - **Fine-tuning details.** For each specialist, we fine-tune the **entire CLIP-based detector** (vision backbone, projection, and classification head), not just the final linear layer. This is detailed in Appendix B.2, and we have clarified the pointer.
> - **Pipeline diagram.** We have added a high-level pipeline diagram in Fig.3, illustrating: (1) specialist training, (2) task-vector construction, (3) SVD into Real core + Fake residuals, and (4) merged-model synthesis.
> - **All-in-one baseline.** The reported All-in-one is trained as a **binary Real vs. Fake classifier** on the union of all specialist data. We also tried a multi-class fake head (Real, FS-Fake, FR-Fake, EFS-Fake) whose fake scores are aggregated; the resulting AUC was quite unstable and sensitive to class frequencies and aggregation, without changing our qualitative conclusions. For this reason, we chose the simpler, more stable binary baseline.
> - **Impact on deepfake detection.** Our focus is complementary to generalization-oriented works. In high-risk applications, once a new forgery family is detected, it is often feasible to collect labeled data and train a specialist. The bottleneck then is **rapid, reliable integration** of that specialist without full retraining. R2M targets exactly this “fast and stable adaptation” regime. Our Uniface (similar FS-style) and MidJourney (highly novel) experiments show that R2M can both (i) handle generators similar to existing families without retraining, and (ii) quickly absorb a newly trained specialist when generalization alone is insufficient. We have clarified this scope and impact in the revised introduction and discussion.

---

### Official Review · Reviewer_2rrG · 2025-11-04

**Soundness:** 3
**Presentation:** 3
**Contribution:** 3
**Rating:** 6
**Confidence:** 3

**Summary:**

This work analyzes the similarity between each specialist detector and the weight‑averaged model in deepfake detection, finding that real features are shared across detectors while fake features differ and can be complementary. Based on this observation, the paper proposes a Real‑aware Residual Model Merging strategy that enables rapid incorporation of new forgery families by preserving the shared real component and merging the residuals from different forgery specialists. Experiments across multiple datasets and protocols demonstrate the effectiveness of the proposed method.

**Strengths:**

1.The proposed method is straightforward and compelling; the analysis of real and fake feature similarities between each specialist and the weight‑averaged model is particularly insightful.
2.The R2M method is novel and effective: it updates models by retaining a shared real component while composing denoised, norm‑matched fake residuals to enable rapid adaptation to new forgeries.

**Weaknesses:**

1.Figure 2’s analysis relies on features from older forgery datasets (e.g., DF40), where real and fake images are relatively easy to separate; how would the analysis and the proposed method perform when forgeries are highly realistic and real and fake feature distributions are not well separated?

**Questions:**

see weaknesses

---

> ### Author Response · Authors · 2025-11-21
> **On Fig. 2 and the use of DF40**
>
> Our goal in this work is not to propose yet another specialist detector that can further separate extremely realistic forgeries from real images, but rather to study **how to merge already well-trained specialists while preserving their performance** after merging. We explicitly focus on the practical and widely encountered regime in which each specialist, regardless of how realistic the underlying forgeries are, has already learned a reasonably good real/fake separation. Our contribution lies in providing a **principled, training-free way to combine such specialists in parameter space**.
>
> Fig. 2 uses DF40 features for clarity of visualization, but the key observation is **relative**, not dataset-specific:
>
> - across multiple forgery families and across different backbones, *real* features from different specialists are highly similar to the weight-averaged model,
> - whereas own-*fake* features remain more complementary.
>
> This pattern is not tied to “easy” forgeries.
>
> Importantly, **DF40 itself spans a broad difficulty spectrum, including highly realistic entire-face synthesis methods (e.g., diffusion/transformer-style generators) in addition to earlier GAN-based forgeries.** Modern detectors trained on these forgeries still rely on subtle cues (frequency-domain artifacts, generator-specific textures, etc.), which are embedded in their parameters.
>
> Our R2M framework is designed to **reuse whatever signals these specialists have already discovered**: we keep their shared real-component while merging only denoised, norm-matched fake residuals in parameter space. As a result, R2M naturally retains the specialists’ robustness even when the forgeries are highly realistic and the distributions are not trivially separable.

---

### Author Response · Authors · 2025-11-21
**General response to all Reviewers**

We would like to sincerely thank all reviewers for their thoughtful feedback and constructive suggestions. The comments greatly helped us improve the clarity and overall quality of the manuscript. We are also encouraged that the reviewers highlighted several strengths of our work, including the principled formulation of real/fake decomposition (Reviewers 2rrG,  3mcr, 9CNL), the strong empirical performance across diverse forgery types and backbones (Reviewers  2rrG, 8SSQ,  3mcr), and the clear writing and well-motivated design of our real-aware residual merging procedure (Reviewers  8SSQ, 9CNL).

We are fully prepared to address all concerns raised and appreciate the opportunity to clarify any remaining questions. Below, we summarize the major revisions made in response to the reviewers’ suggestions, together with pointers to the corresponding sections in the revised manuscript:

1. **Improved Clarity and Expanded Analyses**

     We carefully revised the manuscript to improve clarity in several key sections. In particular, we enhanced the visual explanations in Figs. 2–5 and added accompanying tables in the appendix to ensure that all underlying AUC values are easy to interpret. We also included additional analyses—such as within-category behavior and merging under mixed Real distributions (FF + CDF)—to better illustrate the stability of the shared Real component and the variability of Fake residuals.

2. **Strengthened Experimental Evidence and Robustness Evaluation**

     Responding to the reviewers’ suggestions, we added new experiments covering incremental specialist integration (e.g., Uniface), six-specialist merging, and robustness to highly realistic or previously unseen generators. These results demonstrate that R²M consistently maintains or improves performance while degrading more gracefully than CART in challenging regimes. We also expanded ablations on encoder choices, linear heads, and All-in-one behavior to provide a more complete empirical picture.

3. **Clarified Scope, Assumptions, and Relation to Prior Work**

     We revised the theoretical section to more clearly articulate the local assumptions behind Proposition 1 and specify the scenarios where they naturally hold. In the related-work section, we clarified the distinction between $R^{2}$M and training-time generalization methods such as Effort, emphasizing their complementary roles. To further aid understanding, we added a high-level pipeline diagram illustrating the full merging process.

For detailed, point-by-point clarifications, please refer to our individual responses to each reviewer. If any additional questions arise, we would be glad to provide further explanation during the discussion phase. We appreciate the opportunity to continue refining and improving the paper.

---

### Author Response · Authors · 2025-11-28
**Dear Reviewers**

Dear Reviewers,

We would like to once again express our sincere gratitude for the time and effort you have invested in reviewing our manuscript. As the author–reviewer discussion deadline approaches, we would be deeply appreciative of any further comments or guidance you may wish to provide. We hope that our responses have adequately addressed your concerns, and we would be happy to provide further clarification if needed.

With kind regards,
The Authors

---

### Meta-Review · Area_Chair_gyUh · 2026-01-07

**Summary:**

Summary:
This paper proposes Real-aware Residual Model Merging (R2M), a training-free framework designed to merge multiple deepfake detection specialist models into a single detector. The core idea is to leverage Singular Value Decomposition (SVD) on task vectors to decouple parameters into a shared "Real" subspace (assumed to be consistent across specialists) and generator-specific "Fake" residuals. These components are then processed via rank truncation and norm matching before being recombined. The goal is to enable the rapid integration of new forgery types without retraining.

Strengths:
1. Timely Problem Setting: Applying model merging to deepfake detection is a practical response to the rapid proliferation of generative models, addressing the high cost of continuous retraining.
2. Efficiency: The training-free nature of R2M allows for quick incremental updates (as demonstrated in the Uniface experiment), which is operationally valuable.
3. Intuitive Motivation: The observation that "Real" features are generally shared while "Fake" artifacts are specific is a logical starting point for designing a domain-specific merging strategy.

Weaknesses:
1. Limited Novelty: The conceptual core—using SVD to decouple shared semantic/real content from specific forgery artifacts—bears strong resemblance to prior work, such as "Effort" (ICML'25). While R2M applies this idea at the merging stage rather than the training stage, the underlying mathematical intuition is largely isomorphic, limiting the algorithmic innovation.
2. Theoretical Validity: The theoretical analysis relies on strong assumptions (e.g., local linearity, bounded remainder, mild spectral gap) that typically hold only when models remain very close to their initialization. In realistic deepfake detection scenarios, where specialists require deep fine-tuning to detect subtle artifacts, parameters likely diverge significantly, rendering these assumptions and the resulting propositions unreliable.
3. Robustness to Distribution Shifts: The method relies heavily on the existence of a single "Shared Real Core." This assumption is challenged when real data comes from heterogeneous sources (e.g., different lighting, sensors, or datasets).
4. Security Vulnerabilities: The method explicitly aggregates parameter residuals. This design creates a systemic vulnerability where a poisoned specialist (e.g., with a backdoor in its residual vector) could compromise the merged model, a significant risk in model merging scenarios.

Author Responses: The authors provided a comprehensive rebuttal and added new experiments. However, the results of these experiments ultimately reinforced several key concerns rather than resolving them:
1. Heterogeneous Real Distribution: The authors introduced a new experiment merging specialists trained on different datasets (FF++ and CDF). The results showed a significant performance drop (e.g., AUC falling from ~0.99 to ~0.90) compared to individual specialists. This empirically confirms that the "Shared Real Core" assumption is brittle when the distribution of real data becomes complex, significantly limiting the method's real-world utility.
2. Theoretical Grounding: The authors acknowledged that their theoretical propositions serve more as "heuristic motivation" than global guarantees. Given the highly non-linear nature of deep neural networks, the disconnect between the linear theoretical assumptions and the actual training dynamics remains a major gap.
3. Security: The authors admitted they assume a "trusted training pipeline" and did not offer a technical solution for the security risks associated with merging residuals.
4. Baseline Issues: The "All-in-one" baseline exhibited catastrophic failure (AUC < 0.5) in some settings. While explained as overfitting/label inversion, this suggests the baseline may not have been optimally tuned, potentially inflating the relative gains of the proposed method.

Conclusion:
While the motivation for applying model merging to deepfake detection is sound, the proposed R2M method suffers from significant limitations regarding robustness (particularly with heterogeneous real data), theoretical rigor, and algorithmic novelty. The experimental evidence suggests the method may not generalize well beyond strictly controlled settings. Therefore, the paper is not recommended for acceptance.

**Reviewer Concerns:**

please refer to the summary

**Reviewer Scores:**

please refer to the summary

---

### Decision · Program_Chairs · 2026-01-26

Reject